# Multiplexed live-cell profiling with Raman probes

Chen Chen[1,2], Zhilun Zhao [1,2], Naixin Qian[1], Shixuan Wei[1], Fanghao Hu[1] & Wei Min [1✉]

Single-cell multiparameter measurement has been increasingly recognized as a key technology toward systematic understandings of complex molecular and cellular functions in biological systems. Despite extensive efforts in analytical techniques, it is still generally challenging for existing methods to decipher a large number of phenotypes in a single living cell. Herein we devise a multiplexed Raman probe panel with sharp and mutually resolvable Raman peaks to simultaneously quantify cell surface proteins, endocytosis activities, and metabolic dynamics of an individual live cell. When coupling it to whole-cell spontaneous Raman micro-spectroscopy, we demonstrate the utility of this technique in 14-plexed live-cell profiling and phenotyping under various drug perturbations. In particular, single-cell multiparameter measurement enables powerful clustering, correlation, and network analysis with biological insights. This profiling platform is compatible with live-cell cytometry, of low instrument complexity and capable of highly multiplexed measurement in a robust and straightforward manner, thereby contributing a valuable tool for both basic single-cell biology and translation applications such as high-content cell sorting and drug discovery.

[1] Department of Chemistry, Columbia University, New York, NY, USA. [2]These authors contributed equally: Chen Chen, Zhilun Zhao.
✉email: wm2256@columbia.edu

A comprehensive understanding of fundamental biology, disease mechanisms, and drug therapeutics requires the integration of information from a large number of related pathways. Multiplexed measurements are thus attracting considerable interest in single-cell biology and systems biology[1–3]. In the context of single-cell analysis, mass cytometry is popular for large-scale proteomics[4]. However, mass cytometry is restricted by its destructive nature, and thus it is incompatible with the investigation of live cells and downstream cell sorting. In contrast, fluorescence-based flow cytometry is an optical method that can interrogate live cells non-destructively[5,6]. However, fluorescence detection is fundamentally limited by the broad fluorescent spectra, and the associated "color barrier" constraints the multiplexing level of fluorescence to just a few channels. Although up to 17 parameters have been reported in the literature, the implementation of multiparameter fluorescence cytometry is rather laborious and challenging due to cumbersome and error-prone spectral compensation and high instrument complexity of multiple lasers and detectors[5,7]. Moreover, metabolites are difficult for fluorescence detection because of the relatively bulky size of fluorophores compared to small metabolites[8–10], and as a result, the crucial functional information about metabolic activities is often missing in fluorescence-based live-cell profiling. Furthermore, fluorescence detection often suffers from autofluorescence and photobleaching[11].

Here we exploit Raman scattering as an alternative solution to super-multiplex live-cell profiling. Raman scattering spectroscopy is a powerful optical tool that complements fluorescence in many key aspects. First, in the condensed phase, Raman spectrum is typically 50 times narrower than that of fluorescence. Thus, Raman spectroscopy can principally circumvent the "color barrier" of fluorescence and holds great promise for super-multiplexing[12–15]. Technically, compared to multiple lasers and detectors required in fluorescence cytometry, the single laser (which excites all Raman modes) and single Raman detector (which collects all modes) configuration permits robust readout of multiparameter data with the one-shot acquisition. Second, Raman spectroscopy does not require bulky fluorophores, making it potentially capable of detecting small metabolites[16]. Third, Raman spectroscopy is free from photobleaching.

However, Raman-based live-cell profiling is still in its infancy[17–20]. This is largely because functional Raman probes are significantly lagging behind the counterparts in mass cytometry or fluorescence flow cytometry. In particular, for the important task of profiling specific protein markers, compared to the well-established fluorophore-conjugated antibodies or rare-earth meta-isotope-tagged antibodies[5,21–24], the existing antibody-based Raman probes are limited by insufficient brightness and/or low level of multiplexing, despite extensive efforts in developing organic dyes[25,26], polymer nanoparticles[27,28] and metallic nanoparticles[29,30]. In a sense, Raman spectroscopy cannot fulfill its full potential without the matching functional probes.

Recognizing the lack of suitable functional probes as the bottleneck of Raman-based live-cell profiling technology, herein we devise a panel of multiplexed Raman probe that target a wide range of key molecular and cellular markers. Foremost, by harnessing our recently invented ultra-bright Raman dots (Rdots) and with further functionalization and optimization, we create a panel of Rdots-conjugated antibodies and aptamers, and develop a general method to detect cell surface proteins simultaneously in single live cells. Moreover, by leveraging the independent size and color tunability of Rdots and their good biocompatibility, we succeed in profiling cellular endocytic pathways in a particle-size-dependent manner. Furthermore, multiplexed metabolic activities are profiled in parallel by including small vibrational probes (such as alkyne and C–D bond) with minimal perturbation.

Implemented with a tailored whole-cell spontaneous Raman micro-spectroscopy (Supplementary Movie 1), this live-cell profiling platform offers an innovative strategy to acquire integrated information about cell surface protein abundance, endocytosis activities, and metabolic dynamics simultaneously at a relatively high speed (3600 cells/h) (Fig. 1). We demonstrate the application of this platform in 14-plexed live-cell profiling and phenotyping under various drug perturbations, proving utility in measuring multiparameter information, characterizing cell heterogeneity, revealing underlying correlation, and discovering mechanisms of drug actions. To our best knowledge, this is higher than other Raman-based multiplexing technology of biological targets reported to date. This platform is compatible with live-cell cytometry, of low instrument complexity and capable of highly multiplexed measurement in a robust manner, thereby contributing a valuable tool for both basic single-cell biology and translation applications such as high-content cell sorting and drug discovery.

## Results

**Design and characterization of the expanded panel of Rdots probes.** Despite the high potential for multiplexing, the biological application of Raman spectroscopy is greatly limited by its small Raman cross-section, which is typically $10^8 \sim 10^{14}$ times smaller than that of fluorescent dyes[31]. To amplify the signal, our group has recently developed a simple strategy to prepare ultra-bright Raman dots (Rdots) by non-covalently doping polymer nanoparticles with Raman-active dyes[32]. When coupled with stimulated Raman scattering (SRS) microscopy, these Rdots show a superb imaging performance in immunostaining of intracellular targets. Herein, we aimed to harness ultra-bright Rdots for developing a robust and cost-effective platform for live-cell profiling. For the current application of live-cell profiling, we have tailored Rdots in four aspects. First, unlike imaging intracellular targets where an extremely compact size of Rdots is necessary, the nanoparticle size requirement for profiling living cell surface proteins is much flexible. Therefore, in this work, we prepared Rdots that were 40–120 nm in diameter to further increase their Raman brightness and facilitate the demonstration of endocytosis in a size-dependent manner. Second, we expanded the previously reported 6-color Rdots palette to 10 colors for the need of profiling cell surface proteins and endocytosis. Third, the surface functionalization has also been optimized for this application. Fourth, we moved from immunostaining of fixed cells to profiling living cells.

Briefly, the multi-colored Rdots were prepared via a swelling-shrinking strategy[33]. After the addition of swelling agent THF, adequate Carbow dyes can be incorporated into the swollen polystyrene (PS) beads. Subsequent shrinking and trapping of dyes were achieved by suspension in a large amount of water (Fig. 2a). After doping with Carbow dyes, the Rdots were dispersed evenly without obvious size expansion (Fig. 2b and Supplementary Fig. 1). The Raman peaks of Rdots coincided well with free Carbow dyes in DMSO (Fig. 2c and Supplementary Fig. 2), indicating the maintenance of spectral characteristics after being doped into PS beads. The zeta potential measurements of pure beads and Rdots proved that the surface carboxyl groups were mostly retained after the preparation process (Supplementary Fig. 3), enabling good colloidal stability and compatibility to subsequent bioconjugation. As shown in Fig. 2d, we doped ten Carbow dyes[25] with distinct Raman frequencies into 40, 70, and 120 nm PS beads to prepare ten-colored Rdots, with each Rdot being spectrally resolvable in the Raman-silent window (1800–2600 cm$^{-1}$)[34]. The brightness of Rdots was evaluated through its relative Raman intensity versus EdU (RIE)[35]. By

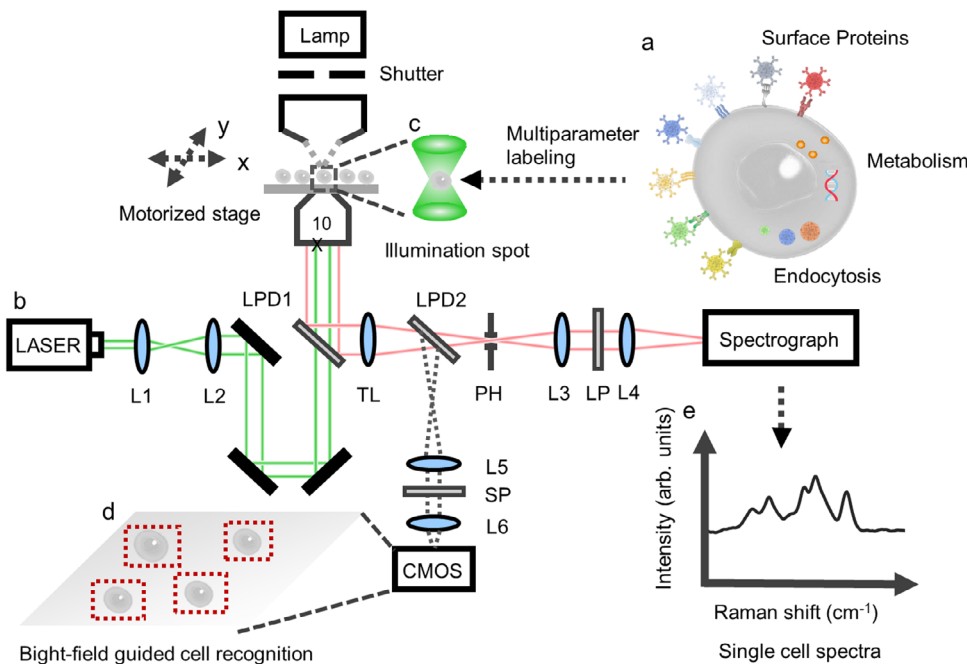

**Fig. 1 Workflow for automated whole-cell multiparameter measurement. a** 14 Raman probes were used to simultaneously quantify cell surface proteins expression levels, endocytosis activities, and metabolic dynamics of individual live cells. **b** A home-built whole-cell confocal Raman micro-spectroscopy can excite and detect all Raman probes simultaneously with a single 532 nm laser and an EMCCD detector. **c** We purposely underfilled the back aperture of the objective to expand the illumination volume (~8 μm in diameter) for the whole-cell excitation. **d** A brightfield image was taken to identify and locate the cells in the field of view. Then, the computer-controlled motorized stage moved the slide so that the illumination spot was parked on the cell. **e** The single-cell Raman spectra were acquired with the automatic confocal micro-Raman system. L1—L6 Lens; LPD1, LPD2 long-pass dichroic beamsplitter; TL tube lens; PH pinhole; LP long-pass filter; SP short-pass filter.

virtue of the large Raman cross-sections and high local concentration of Carbow dyes enriched inside, most Rdots exhibit tremendously high RIE values over $10^5$ (Fig. 2e), orders of magnitudes higher than the organic Raman probes developed recently[25,36]. As a result, a nanoparticle-based Raman color palette was achieved in Fig. 2d with highly multiplexing capability and ultra-brightness.

**Live-cell profiling of surface proteins of single cells by Rdots conjugates.** To profile living cell surface proteins, we then functionalized prepared Rdots with antibodies or aptamers. A non-toxic and hydrophilic polymer polyethylene glycol (PEG) was employed to reduce potential non-specific hydrophobic interaction and electrical adsorption of Rdots onto cell membrane[37]. As shown in Fig. 3a, the surface of 40 nm carboxylated Rdots were functionalized with long 5 kDa amine-PEG-acid and backfilled with short 1 kDa amine-PEG-alcohol to create a cage-like shell with EDC/NHS coupling chemistry. The DLS characterization of Rdots conjugates was demonstrated in Supplementary Fig. 4. The surface coverage of the PEG layer minimizes the non-specific binding and supports a flexible spacer arm for an oriented bio-conjugation to facilitate targeting efficiency. For protein recognition, the surface amine-PEG-acid was then covalently conjugated with targeting antibodies or aptamer with EDC/NHS chemistry.

In our previous work, Rdots were imaged by narrowband SRS microscope in which only one Raman channel can be imaged at a time[32]. In this work, we aimed to acquire single-cell Raman spectra of all vibrational modes in a cost-effective manner by employing whole-cell spontaneous Raman spectroscopy[38]. Here, the illumination laser spot was intentionally expanded to ~8 μm to ensure the excitation of the whole mammalian cell. By doing

so, the Raman signal over the entire cell volume can be optically integrated as an individual spectrum with rapid acquisition of multiplexed probes, suitable for high-throughput single-cell applications (Fig. 1). To test the feasibility of the detection, PEGylated Rdots were firstly conjugated with a primary antibody against CD44, a cell surface adhesion molecule that is frequently overexpressed on cancer cells[39]. Rdot2218-CD44 stained HeLa cells exhibit a single, narrow, and strong Raman peak in the cell-silent window under this setup (Fig. 3b), thanks to the ultra-brightness of Rdots and optical integration of all Rdots labeled to single cells. Figure 3c shows the corresponding SRS image at 2218 cm$^{-1}$, and the membrane-bound SRS signal confirms the subcellular localization of CD44 receptors and validates that Rdots recognition was specific enough for the detection of membrane proteins. The recognition specificity of Rdots conjugates was further validated in Supplementary Figs. 5 and 6.

As a proof of concept of two-channel cell profiling, we assessed HeLa and SKBR3 cell lines stained with both Rdot2153-CD55 and Rdot2079-CD44. CD44 and CD55 are both overexpressed in HeLa cells, but only moderately expressed in SKBR3 cells. Single-cell spectra were acquired through our whole-cell Raman micro-spectroscopy (Fig. 3d). As expected, HeLa cell exhibits a higher Raman intensity for both CD44 and CD55 proteins, which correlates well with the expression level of surface proteins (Fig. 3e). A two-dimensional scatter plot of individual cells, based on the quantitative readouts of two protein channels, present two clearly separable clusters with certain spread each (Fig. 3f), which not only indicates distinct surface receptor profiles between HeLa and SKBR3 but also suggests expression heterogeneity of individual cells within each cell type.

We then sought to simultaneously profile multiple cell surface proteins on SKBR3 with a cocktail of functionalized Rdots probe. Seven-colored 40 nm Rdots were conjugated with recognition

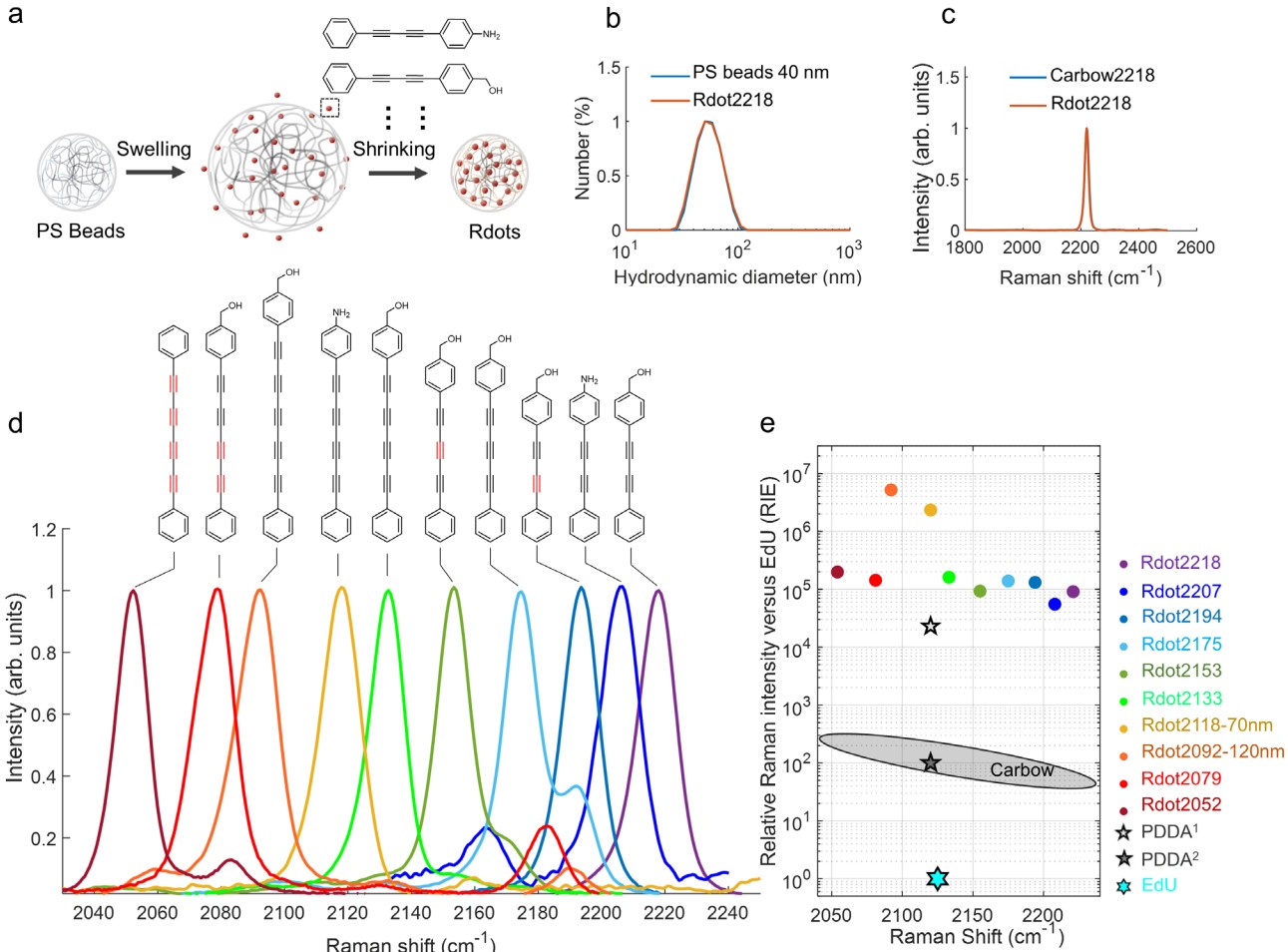

**Fig. 2 Preparation and physical properties of ten-colored Rdots. a** Schematic illustration of Rdots preparation via swelling-shrinking strategy. **b** Dynamic light scattering (DLS) size characterization of Rdot2218 before and after dye doped. **c** Raman spectra of Rdot2218 in water and free Carbow2218 dye in DMSO. **d** Normalized spontaneous Raman spectra of ten-colored Rdots, with corresponding doped dye structure listed above. **e** RIE values and Raman peak positions of ten-colored Rdots. Most Rdots exhibit tremendously high RIE values over $10^5$, at least 5 times larger than PDDA[36]. Due to the volume effect, Rdots with larger sizes (70 nm and 120 nm) have an RIE value over $10^6$. RIE values were measured with a spontaneous Raman spectrometer with 532 nm excitation. PDDA[1] is the RIE value detected under the resonance Raman effect. PDDA[2] is the RIE value detected without resonance enhancement.

molecules (primary antibodies or aptamers) against nucleolin (Rdot2218-Nucleolin), EpCAM (Rdot2194-EpCAM), MUC1 (Rdot2175-MUC1), CD55 (Rdot2153-CD55), EGFR (Rdot2133-EGFR), CD44 (Rdot2079-CD44), and HER2 (Rdot2052-HER2), respectively. Among them, nucleolin, EpCAM, MUC1, and HER2 were stained via aptamers and the others were by antibodies. Firstly, the specificity and semi-quantitative measurement of Rdots probes were individually confirmed (Supplementary Fig. 7), where the Raman signals were found to be correlated with the expression level of surface proteins. After these individual validations, we then pooled the seven functionalized Rdots into a cocktail and stained the corresponding surface proteins simultaneously. Seven resolvable Raman peaks were clearly observed, and could be assigned to seven surface proteins, respectively (Fig. 3g, zoom-in view in Fig. 3h). Thus, we have created a panel of Rdots-functionalized antibodies and aptamers to profile multiplex surface proteins on live cells.

**Endocytic pathway profiling by Rdots of various sizes and colors.** Upon exposure to mammalian cells, PS beads have been reported to enter cells through endocytosis, subsequently reaching the endosomal components[40]. Although some of the

endocytic pathways have been well established, their exact biological roles are still under investigation[41,42]. Intriguingly, it has been suggested that the mechanisms and pathways by which the nanoparticles were internalized are strongly dependent on the size of particles[43]. Motivated by the independent size and color tunability of Rdots, we envisioned to profile cellular endocytic pathways by incubating cells with Rdots of different sizes and colors. Herein, Rdots of 40 nm (Rdot2207), 70 nm (Rdot2118), and 120 nm (Rdot2092) in diameter were incubated with HeLa cells, and their Raman peaks are designed to be mutually resolvable. Indeed, after 6 h, HeLa cells exhibit three resolvable Raman peaks in the cell-silent window, indicating cellular uptake for each Rdots (Fig. 4a). The spatial distribution of three endocytic Rdots visualized by SRS indicates that Rdots of different sizes all tend to accumulate in the perinuclear region (Fig. 4b–d and Supplementary Fig. 8).

For whole-cell Raman acquisition, the corresponding Raman intensity is proportional to the number of internalized Rdots, since the Raman intensity of Rdots remains stable after exposed to cellular microenvironments (Supplementary Fig. 9). We, therefore, quantified the peak intensity to profile the differential mechanism of Rdots internalization. To this end, cells were first pre-treated with one of three inhibitors, chlorpromazine (CPZ),

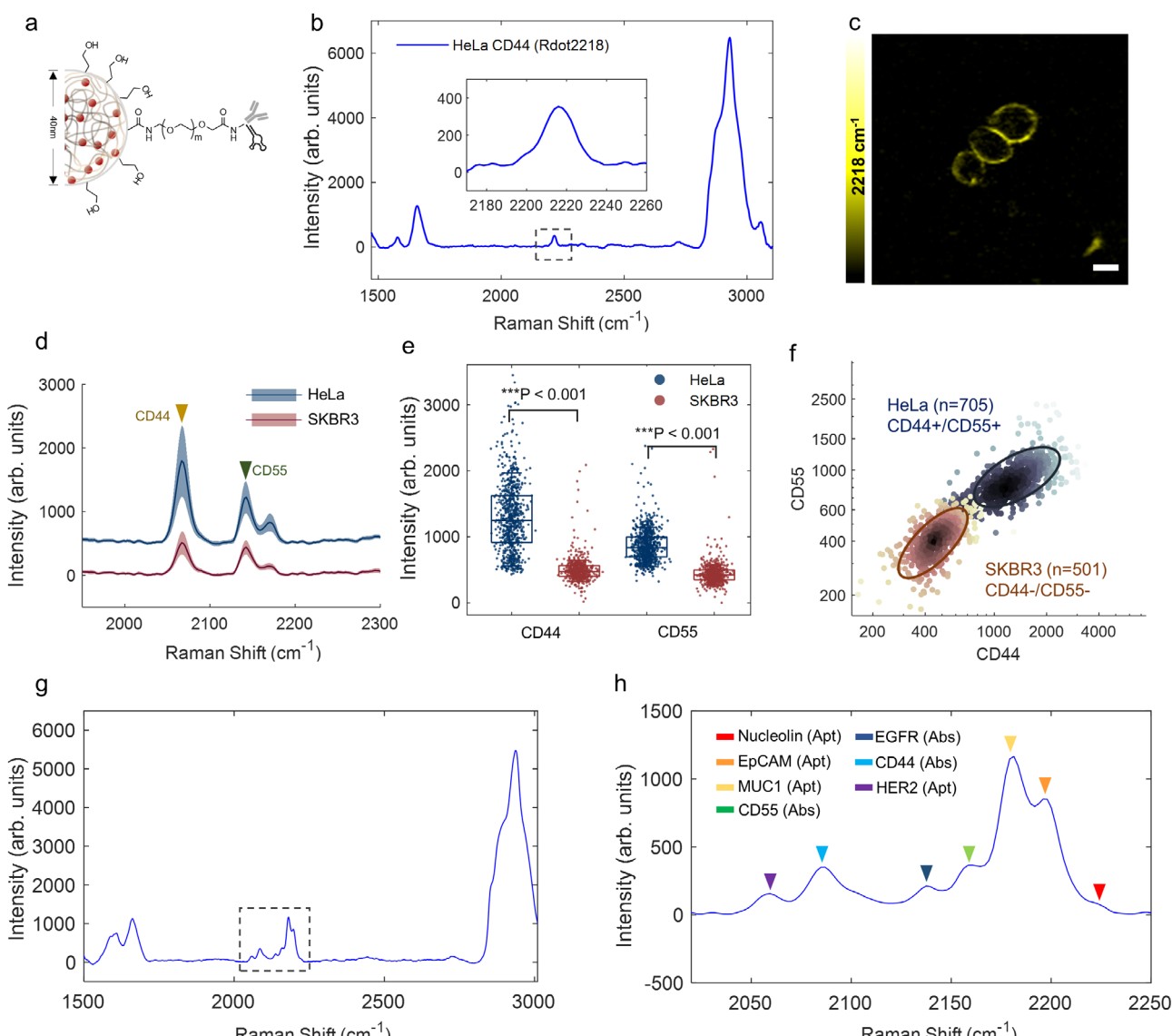

**Fig. 3 Rdots-conjugated antibodies and aptamers for live-cell profiling of surface proteins of single cells. a** Schematic structure of Rdots conjugates. Carboxylated Rdots were first attached with amine-PEG-alcohol and amine-PEG-acid at a molar ratio of 9:1. The PEGylated Rdots were then conjugated with targeting molecules (antibody or aptamer) for recognition. **b** Spontaneous Raman spectra of Rdot2218-CD44 stained HeLa cell, averaged over 100 individual cells. **c** SRS image of Rdot2218 positively staining of CD44 on HeLa cell surface, scale bar: 10 µm. The color scales display relative SRS signal levels. Image results were representative of three biological replicates. **d** The averaged spontaneous Raman spectra of HeLa and SKBR3 cells acquired after dual-color staining. Solid lines indicate the averaged Raman spectra, the shaded area indicates the standard deviation of Raman spectra from multiple cells ($N_{HeLa} = 705$; $N_{SKBR3} = 501$). **e** Box plot illustrating the Raman intensity of CD44-targeted Rdot2079 and CD55-targeted Rdot2153 on HeLa and SKBR3 cell surface. Box edges indicate the interquartile range (IQR); center line, median; whiskers, lowest and highest data within 0.5IQR from the first and third quartiles, respectively. *** denotes a significant difference between the two groups ($P < 0.001$, two-tailed). $N_{HeLa} = 705$; $N_{SKBR3} = 501$. The exact $P$ values were provided in the source data. **f** Two-dimensional scatter plot of individual cells, based on the quantitative readouts of two protein channels. **g**, **h** Spontaneous Raman spectra of SKBR3 cells acquired after stained with seven Rdots conjugates, averaged over 600 individual cells (**g**), and zoom-in view (**h**).

nystatin, and cytochalasin D (Cyto D). CPZ is a cationic amphiphilic drug used to block clathrin-mediated endocytosis[44]. Nystatin can decompose cholesterol and inhibit caveolae-mediated endocytosis[45]. Cyto D inhibits the polymerization of actin filaments, which is required for receptor-mediated endocytosis in mammalian cells[46]. Additionally, cells were incubated at 4 °C to inhibit energy-dependent endocytosis. Then endocytosis was examined by the quantification of internalized Rdots at the single-cell level. A distinctive response of Rdots internalization was detected when treated with various inhibitors (Fig. 4e–g), and the radar plot was presented to highlight the fold change after

inhibition (Fig. 4h). A similar decreased pattern was observed for Rdots uptake by cells pre-treated with CPZ, implying that their entry into the cells is sensitive to the assembly of clathrin-coated pits[47]. In contrast, for nystatin-treated cells, the uptake of Rdots decreased as size increases. The internalization of 40 nm Rdots was unaltered or even slightly higher, while 70 and 120 nm Rdots were both inhibited, indicating that caveolae receptors are preferentially involved in the internalization of larger Rdots[48,49]. Consistent with blocked endocytosis at low temperature[49], exposure of cells to a low temperature significantly reduced the uptake of all three Rdots. Rdots internalization is also suppressed

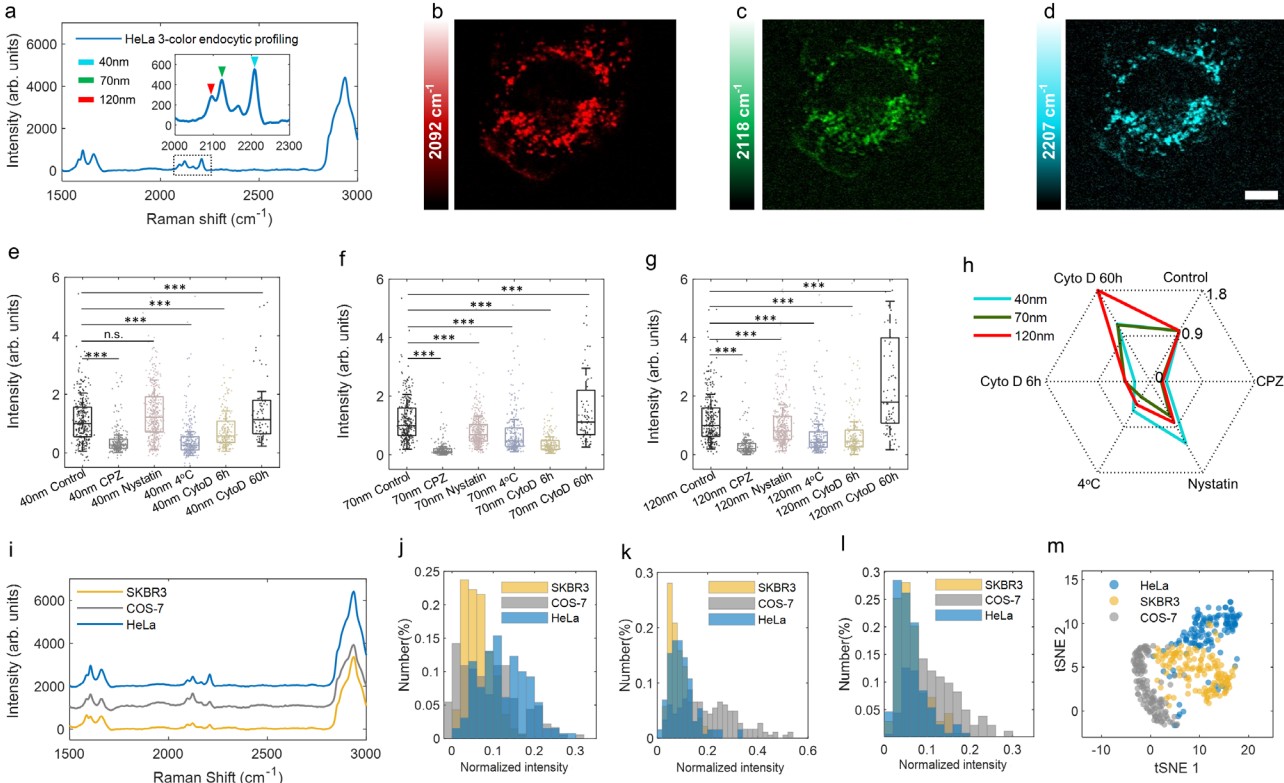

**Fig. 4 Endocytic profiling assay by multi-color Rdots. a** Spontaneous Raman spectra of HeLa cells acquired after incubated with Rdot2207 (40 nm), Rdot2118 (70 nm), and Rdot2092 (120 nm) for 6 h, averaged over 128 individual cells. The peak position of each Rdots was indicated by the arrow. **b–d** SRS images of cellular distribution for three endocytic Rdots, **b** 40 nm, **c** 70 nm, **d** 120 nm, scale bar: 10 μm. The color scales display relative SRS signal levels. Image results were representative of four biological replicates. **e–g** Box plot illustrating the normalized Raman intensity distribution of three endocytic Rdots under the stress of clathrin inhibitor CPZ and caveolae inhibitor nystatin, low temperature and actin inhibitor Cyto D (incubated for 6 h and 60 h), **e** 40 nm, **f** 70 nm, **g** 120 nm. Box edges indicate the interquartile range (IQR); center line, median; whiskers, lowest and highest data within 0.5IQR from the first and third quartiles, respectively. $N_{control} = 309$; $N_{CPZ} = 209$; $N_{nystatin} = 387$; $N_{4°C} = 255$; $N_{Cyto D 6h} = 198$; $N_{Cyto D 60h} = 87$. *** denotes a significant difference between the two groups ($P < 0.001$, two-tailed). n.s. denotes not significant ($P > 0.05$, two-tailed). The exact $P$ values were provided in the source data. **h** Properties of inhibition displaying on Radar diagrams and each inhibition action is provided with an axis. **i** Averaged spontaneous Raman spectra of HeLa, SKBR3, and COS-7 cells acquired after incubated with three endocytic Rdots for 6 h, $N_{HeLa} = 128$; $N_{SKBR3} = 134$; $N_{COS-7} = 130$. **j–l** Histogram showing the Raman intensity distribution of three endocytic Rdots uptake by HeLa, SKBR3, and COS-7 cells, **j** 40 nm, **k** 70 nm, **l** 120 nm. **m** Three-color endocytic profiling of HeLa and SKBR3 and COS-7 cells projected on t-SNE plot, colored by cell types.

by short-term Cyto D incubation, implying that actin filaments are necessary for efficient endocytosis. Interestingly, after incubation with Cyto D for 60 h, the cellular uptake of Rdots was markedly promoted, likely due to apoptosis-induced membrane permeability alteration. Therefore, the results of inhibition treatments suggest that different particle sizes do differ during endocytosis.

We then demonstrate this method in investigating endocytosis differences in HeLa, SKBR3, and COS-7 cell lines. After pre-treated with three-color/size endocytic Rdots for 6 h, single-cell Raman spectra were acquired. The spectral feature for three cell lines appears distinct in the cell-silent region, suggesting different endocytic performances (Fig. 4i). Quantitative readouts of three endocytic Rdots were then analyzed and histograms were constructed from single cells (Fig. 4j–l). We note that histogram from a single-type Rdot (either 40 or 70 or 120 nm) alone has less discriminate power to distinguish between HeLa, SKBR3, and COS-7 cells, largely because of the wide distributions from cell-to-cell variation. To harness information from all three endocytic Rdots, a multiparameter-based t-distributed stochastic neighbor embedding (t-SNE)[50] technique was employed, and three clusters were identified (Fig. 4m), suggesting that endocytosis could serve as an informative dimension of cell phenotyping. This success showcased the great potential of endocytic profiling in cell-type

identification, as well as the advantage of employing endocytic Rdots of different sizes.

**Development of 14-plexed live-cell Raman profiling platform.** As shown above, we have successfully developed seven-colored Rdots to evaluate cell surface proteins (nucleolin, EpCAM, MUC1, CD55, EGFR, CD44, and HER2) and three-colored Rdots (Rdot2207, Rdot2118, and Rdot2092) to probe cellular endocytosis in live cells. No significant signs of cytotoxicity were observed during Rdots staining and endocytic test, supporting their feasibility in the live-cell assay (Supplementary Fig. 10). To provide a more comprehensive picture of phenotypic diversity of single cells, we then incorporated four additional metabolic probes, $^{13}C$-EdU, 17-Octadecynoic Acid (17-ODYA), diyne-tagged CoQ analogs AltQ2[35] and $^{13}C$-amino acids ($^{13}C$-AA), to enrich the Raman probe panel (Fig. 5a). Under the substitution of $^{13}C$-AA, the Raman peak at 1574 cm$^{-1}$ and 1659 cm$^{-1}$, ascribed to cellular amide II and amide I, will shift to lower wavenumber by about 40 cm$^{-1}$, as a result of the $^{13}C$ containing amide bonds in the newly synthesized protein[51]. The metabolic dynamics of fatty acids and DNA synthesis were probed by the alkyne-tagged fatty acid 17-ODYA and $^{13}C$-EdU[52,53], respectively. The electron transport in the mitochondria respiratory chain was characterized

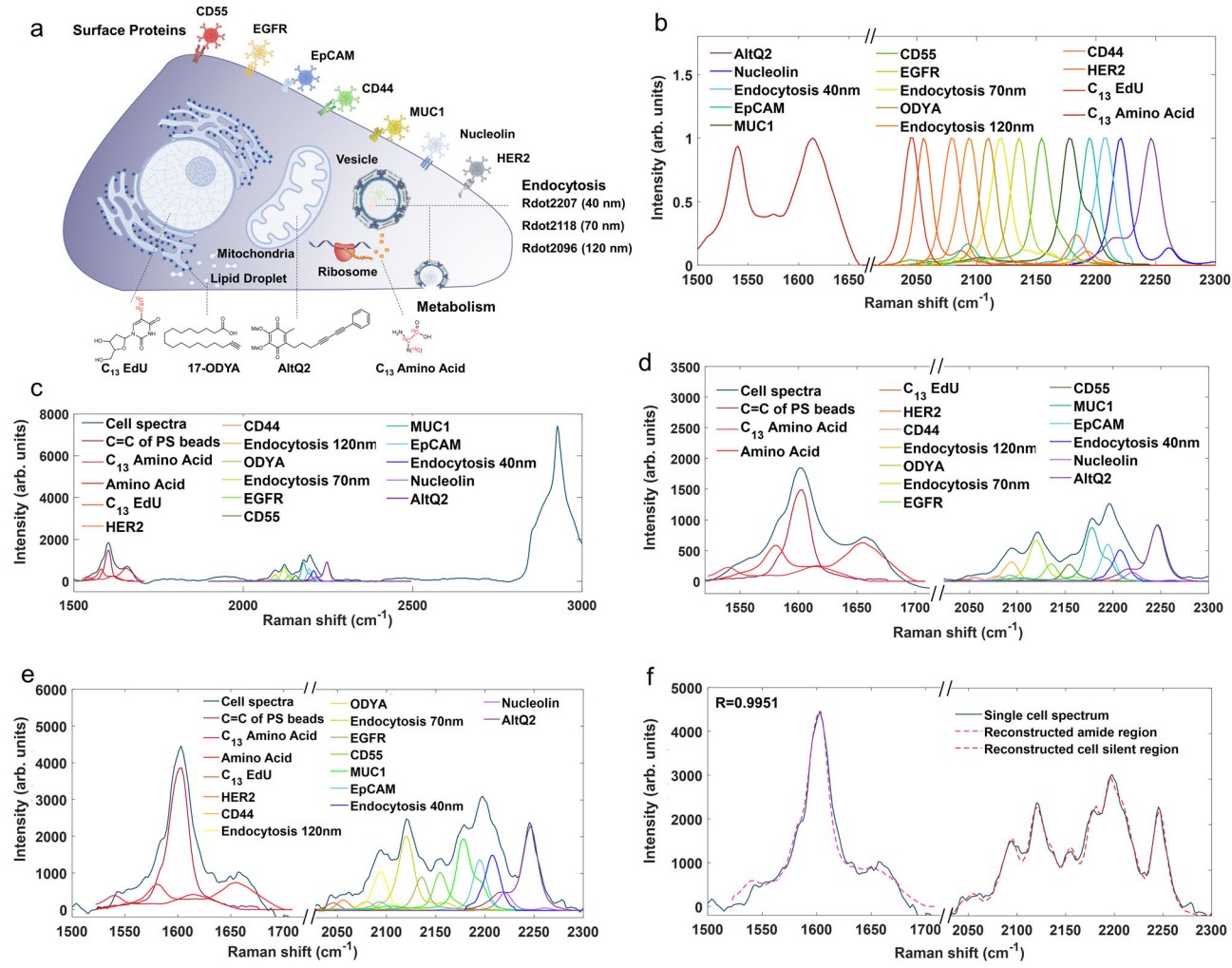

**Fig. 5 Principle and implementation of multiparameter Raman profiling of individual living cells. a** Diagram illustrating the cellular localization of 14 Raman probes for multiparameter live-cell profiling. **b** Normalized Raman peaks of 14 Raman probes with distinct frequencies. **c, d** 14-plexed live-cell Raman spectra with unmixing processing (**c**) and zoom-in view of cell amide and silent window (**d**), averaged over 213 individual cells. **e, f** Zoom-in view of single live-cell Raman spectrum with unmixing processing (**e**) and reconstructed amide and cell-silent region (**f**), the Pearson correlation efficient between reconstructed and original spectra $R = 0.9951$.

by CoQ analogs AltQ2[35]. These metabolic activities of small metabolites are difficult for fluorescence methods[16]. In total, 14 Raman probes were included in our multiplexed panel (Fig. 5b).

Notably, almost all of the 14 probes show individual narrow Raman peaks at the cell-silent window, enabling highly sensitive detection in a background-free manner. Meanwhile, we designed them so that their Raman peaks can be mutually separated from each other. In contrast, such a resolvability would be challenging for fluorescence microscopy with broad emission peaks. To acquire the 14-plex Raman information altogether, we implemented a home-built whole-cell confocal Raman microspectroscopy which can excite and detect all Raman probes simultaneously with a single 532 nm laser source and an EMCCD detector[38]. Equipped with fully automated hardware and brightfield-guided cell identification and localization algorithms (Supplementary Fig. 11), efficient acquisition was achieved for single-cell readouts with a throughput of 3600 cells per hour (1 s per cell). Compared with fluorescence-based flow cytometry which requires multiple lasers and detectors and cumbersome spectral compensation, the Raman instrument here is straightforward, reliable, and cost-effective.

We then demonstrated our 14-plex Raman probe profiling cocktail under whole-cell Raman micro-spectroscopy. Metabolic probes including ${}^{13}$C-EdU, 17-ODYA, and ${}^{13}$C-AA were first supplemented to cell culture media for 60 h to report various metabolic activities. For endocytic profiling, cells were subsequently incubated with three Rdots for 6 h. Then, seven functionalized Rdots cocktail was pooled to stain cell surface proteins simultaneously. Prior to the spectral acquisition, AltQ2 probe was added to live cells (see "Methods" section for details). Single-cell Raman spectra of SKBR3 cells were acquired with 1 s acquisition time (Fig. 5c and Supplementary Fig. 12). To decipher the contribution of individual Raman probes, the amide region (1500–1700 cm$^{-1}$) and cell-silent region (2000–2300 cm$^{-1}$) were linearly decomposed by using the reference spectral profiles of each probe (Fig. 5d)[54–56]. We applied the unmixing algorithm on single-cell Raman spectra (Fig. 5e), and the spectrum reconstructed from unmixed components is highly consistent with the single-cell spectrum measured originally (Fig. 5f). The decomposed peak intensity for the Raman probe with relatively low SNR still coincides well with the ground truth value, indicating a reliable retrieval after unmixing (Supplementary Fig. 13). Taken together, a robust Raman-based platform is established to provide a highly multiplexed profiling of live cells. This is higher than other Raman-based multiplexing technology of biological targets so far, going beyond the previous record of

8-plex tissue imaging and 10-color live-cell imaging in the literature[25,26].

**Live-cell multiparameter profiling differentiates various drug actions.** Cell-based profiling techniques are increasingly being used in drug discovery to monitor cell response upon drug treatment[57]. Optical techniques have distinctive advantages by allowing non-invasive, fast, and cost-effective readouts of individual living cells[58]. However, most high-content profiling assays have been primarily focusing on partial aspects of biological phenotypes[59]. Integration of different phenotypes will facilitate a more systematic understanding of the mechanism of drug candidates[60]. To this end, we demonstrate our live-cell multiparameter profiling platform in revealing phenotypic responses upon five chemotherapy reagents (Fig. 6a). Trastuzumab is a HER2-targeted monoclonal antibody developed for clinical use in breast cancer patients[61]. Hydrogen peroxide ($H_2O_2$) is well known as a major member of reactive oxygen species (ROS)[62]. Cycloheximide (CHX) exerts its action by inhibiting protein synthesis[63]. Cyto D causes the disruption of actin filaments and inhibition of actin polymerization[64]. Cisplatin is an inhibitor of DNA synthesis and cell growth[65]. After labeling with 14 Raman probes, single-cell Raman spectra of SKBR3 cells under treatment of five chemotherapy reagents, as well as untreated cells, were acquired through the automated whole-cell Raman microspectroscopy (Fig. 6b).

We then decomposed single-cell Raman spectra to retrieve the intensities of 14 individual probes and constructed a population-averaged heatmap for all 14 phenotypic features (Fig. 6c and Supplementary Fig. 14). Our results are largely consistent with literature reports. For instance, Cyto D treatment inhibits both protein and DNA synthesis, consistent with previous findings[66,67]. Long-term Cyto D treatment will promote cellular internalization of Rdots, consistent with our demonstration above (Fig. 4h). Previous research has shown that inhibition of actin polymerization, such as by Cyto D, suppresses CD44 surface expression[68], which is also agreed with our profiling results, further proving the reliability and capability of our drug action study. CHX can interfere with the process other than protein synthesis such as DNA synthesis by multiple direct and indirect mechanisms[69], which has been captured by our $^{13}C$-EdU and $^{13}C$-AA probes. For $H_2O_2$-treated cells, the intermediate ROS generated by $H_2O_2$ is potent oxidants of proteins, lipid, and nucleic acids, thus retarding their turnover rate[70,71], consistent with the phenotype revealed by 17-ODYA. Our results also revealed some fresh insights. For example, cellular internalization of Rdots of 70 and 120 nm was also promoted under $H_2O_2$ treatment, indicating a membrane permeability change. For another example, increased internalization of AltQ2 was observed after treated with Cyto D, CHX, and $H_2O_2$, indicating alteration of mitochondrial membrane potential by these chemotherapeutic drugs[72–75].

**Single-cell measurement enables clustering, correlation, and network analysis.** One transformative advantage of single-cell measurement over the conventional population-averaged one is the detailed information from a large number of individual cells. This is especially important for highly multiplexed readouts, as extensively documented in single-cell transcriptomics[76–78]. Our single-cell data also allowed us to perform clustering, correlation, and network analysis, which are all beyond the population-averaged results. For clustering analysis, a two-dimensional t-SNE plot was constructed to process the high-dimensional Raman spectral readouts from all the cells (Fig. 6d). It is intriguing that cells treated by different chemotherapy reagents form clear clusters in the t-SNE space, highlighting distinct phenotypic responses.

Pearson correlation coefficients can be computed between any pairs of 14 phenotypic features across all the measured cells (Fig. 6e). Two tumor-associated proteins MUC1 and EpCAM exhibit a highly positive correlation, thereby suggesting a consistent role in cancer metastatic progression[79]. Interestingly, a strong correlation was observed over the internalization of three endocytic Rdots, likely due to that they are probing the same class of biological process. The expression of transmembrane receptor EGFR is positively correlated with endocytic channels, which is reasonable since endocytosis plays an important role in EGFR-mediated cell signaling[80].

We then analyzed the correlation network of 14 phenotypic features to reveal a systematic co-regulation and interaction across cell metabolites, endocytosis, and surface proteins (Fig. 6f–k). For untreated cells, HER2, EpCAM, EGFR, and 120 nm endocytosis are associated with the most other phenotypic features, thereby acting as central nodes of the network (Fig. 6f). According to the "centrality-lethality rule", these nodes tend to be essential in biological networks and more likely to be identified as treatment targets[81]. In contrast, metabolic probes ODYA and $^{13}C$-EdU locate away from the primary cluster, suggesting that those nodes might work independently. Analysis of the drug-induced network topology can facilitate understanding of drug responses and accelerate drug development[82]. Interestingly, different network topologies emerged over drug perturbations. For example, after CHX treatment, metabolites nodes tend to approach the cluster center (Fig. 6h), suggesting metabolites form more efficient feedbacks to drive cells returning to normal state[83]. $H_2O_2$-treated cells exhibit two neighboring network modules connected by the bridge nodes (CD55 and EpCAM, Fig. 6k). The independent regulation of both modules from those bridge nodes makes them attractive as drug targets[83]. This observation can further help to explore the appropriate drug combinations and multi-target drugs[84]. Taken together, the rich information extracted from single-cell multiparameter profiling holds great potential in unraveling complex interactions between multiple molecules for predicting uncertain drug mechanisms.

## Discussion
By virtue of its distinctive spectroscopy, Raman scattering holds great potential for providing a powerful technology towards super-multiplexed live-cell profiling with a large number of phenotypes. However, its current performance is largely hindered by the lack of functional probes targeting specific molecular and cellular markers. On one hand, this under-development is likely a consequence of the conventional label-free paradigm in the Raman research field; on the other hand, it is partly due to the technical difficulty of creating Raman probes that are sufficiently bright and of many colors. In this study, we addressed this pressing need and devised a multiplexed probe panel to simultaneously quantify cell surface proteins, endocytosis activities, and metabolic dynamics of an individual live cell. Inspired by our recent design of ultra-bright Rdots, we further developed a set of Rdots with different sizes, surface functionalization, and color display. Their RIE value can reach $4 \times 10^7$, making them the brightest organic-based Raman probes. We functionalized these Rdots with antibodies and oligonucleotide aptamers into viable probes for profiling multiple cell surface proteins at the single-cell level. Further benefiting from the good biocompatibility and size tunability of Rdots, we developed a method for probing the endocytic pathway. Meanwhile, metabolic activities are simultaneously profiled by employing vibrationally tagged metabolites. The combination of these three categories of probes with proper

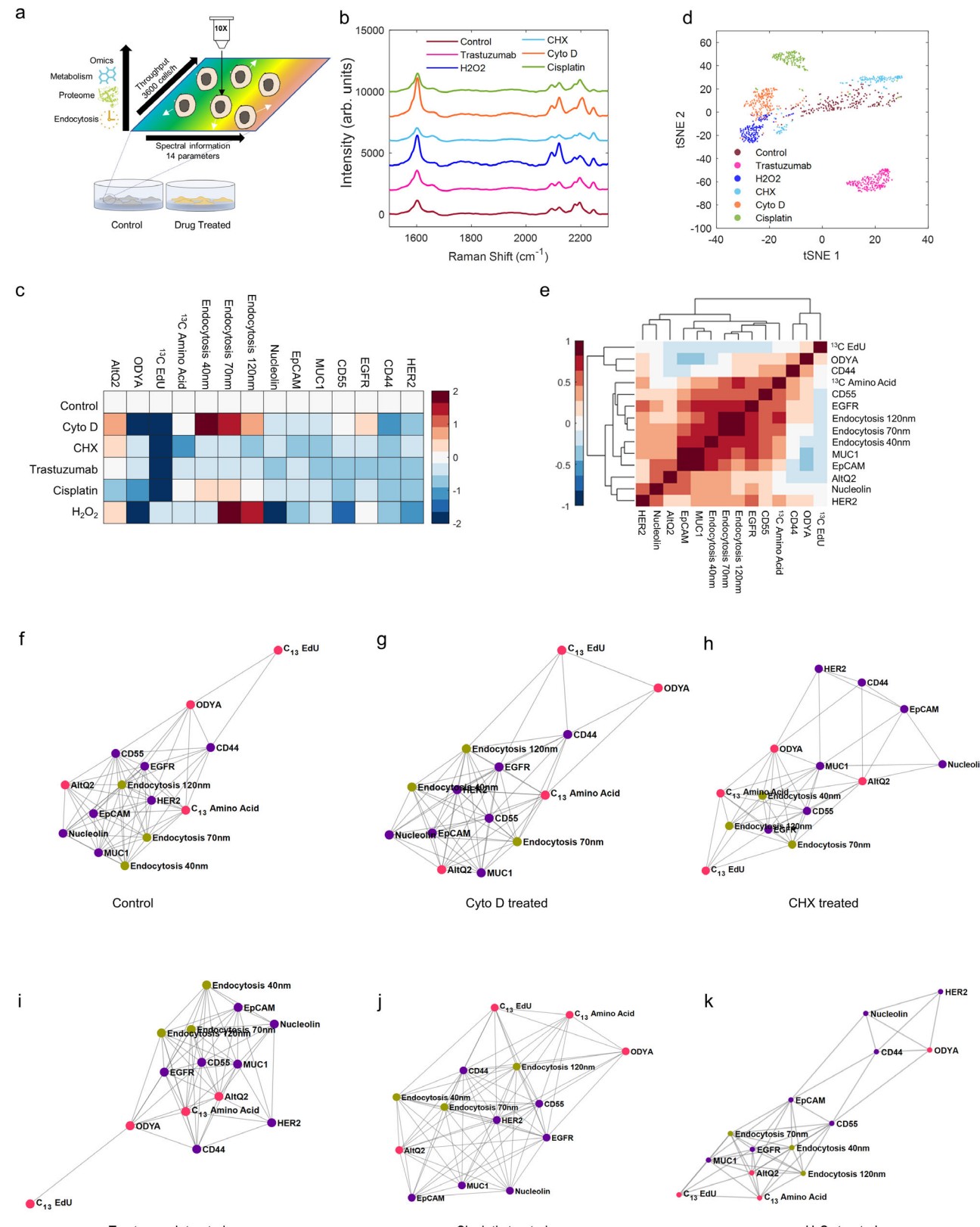

color arrangement generates 14-plexed Raman probes (Fig. 5b), higher than other Raman-based multiplexing reported so far. Coupling with automated whole-cell Raman micro-spectroscopy provides a robust and cost-effective platform to profile comprehensive phenotypes of a single live cell. We demonstrate the utility of this platform in revealing cellular response

characteristics under different drugs, from both the population-averaged analysis (such as heatmap) and the single-cell analysis (including clustering, correlation, and network).

Compared to the well-established mass cytometry and fluorescence-activated cell sorting, the presented technique exhibits a few notable advantages: it is compatible with live-cell

**Fig. 6 Live-cell multiparameter profiling reveals cellular response characteristics for different drugs. a** Schematic illustration of phenotypic profiling over multiple chemotherapy treatments. **b** Averaged spontaneous Raman spectra of SKBR3 cells acquired after drug perturbation and probes incubation, $N_{Control} = 175$; $N_{CytoD} = 182$; $N_{CHX} = 133$; $N_{trastuzumab} = 246$; $N_{Cisplatin} = 131$; $N_{H2O2} = 145$. **c** The population-averaged heatmap for individual parameters reveals the mechanisms of multiple drug actions. The intensity of each parameter is normalized to the control sample. Dark red denotes increased intensity, dark blue denotes decreased intensity. **d** t-SNE scatter plot clustered SKBR3 cell populations based on multiparameter profiling results, colored by drug treatments. **e** Unsupervised hierarchical clustering heatmap analysis of Pearson correlation coefficients over 14 parameters for untreated control cells. Dark red denotes high positive correlations, dark blue denotes high negative correlations. **f–k** Network graphs of correlations among 14 parameters for untreated control cells (**f**); Cyto D-treated cells (**g**); CHX-treated cells (**h**); Trastuzumab-treated cells (**i**); Cisplatin-treated cells (**j**); H$_2$O$_2$-treated cells (**k**). All nodes are connected and the length of the edges connecting two nodes represents the inverse of correlation degree between two parameters.

cytometry, of low instrument complexity, and capable of highly multiplexed measurement in a robust and straightforward manner. Moreover, vibrationally tagged metabolites can be readily included to report metabolic activities, which makes an important contribution to metabolomics profiling but is difficult for other methods. Further improvements to this technique are possible. Along the direction of probe development, the color palette of Rdots can be further expanded by incorporating more Raman-active probes beyond Carbow. We expect the level of multiplexing will continue to grow with the expanded Rdots panel. Along the direction of instrumentation, although we only employed spontaneous Raman micro-spectroscopy to demonstrate the cost-effectiveness, it should be straightforward to implement on more advanced setups such as coherent Raman microscopy including CARS and SRS[10,17,85]. By virtue of nonlinear amplification, as well as the quick parallel detection[85] and scanning[17] systems, SRS microscope can enable real-time SRS image of single live cells with a throughput of up to ~140 events per second. Moreover, a flow setup equipped with sorting feature[18,86] would facilitate downstream cell sorting. Given the existing performance and expected improvements, we believe this live-cell-based profiling approach holds great potential for both basic single-cell biology and translation applications such as high-content cell sorting and drug discovery.

## Methods
A step-by-step protocol can be found at Protocol Exchange[87].

**Cell culture**. HeLa, SKBR3, and COS-7 cells were all obtained from the American Type Culture Collection (ATCC) and kept under standard cell culture conditions (5% CO$_2$, 37 °C). HeLa and COS-7 cells were cultured in DMEM media (Gibco, 11965118) supplemented with 10% fetal bovine serum (FBS, Gibco, 10099141) and 1% penicillin/streptomycin (P/S, Gibco, 15140148). SKBR3 cells were cultured in McCoy's 5A media (Gibco, 16600082) supplemented with 10% FBS and 1% P/S.

**Rdots preparation and bioconjugation**. Incorporation of Carbow dyes was achieved by swelling the 4% w/v polystyrene (PS) beads (Invitrogen, C37232, C37233, C37479) in a solvent mixture containing 160 μL 4% w/v PS beads, 160 μL reverse osmosis (RO) water, and 120 μL Tetrahydrofuran (THF, Sigma, 401757), and by adding a controlled amount of Carbow dyes to the mixture (refer to Supplementary Table 1 for specific dye concentration). After 30 min of gentle agitation at room temperature (RT), 2 mL 20 mM phosphate buffer (pH 7.3) was subsequently added to shrink the Rdots. Excess dyes were removed by three rounds of centrifugation and resuspension in RO water using 30K MWCO filters (Millipore, UFC9030). Rdots bioconjugation to antibodies and aptamers were carried out through carboxyl-to-amine crosslinking using the ethyl dimethylaminopropyl carbodiimide (EDC, Thermo Scientific, 22980) and sulfo-NHS (Sigma, 56485). To activate the carboxyl groups on beads surface for covalent bond, 200 μL 4% w/v beads were mixed vigorously with 100 μL 100 mg/mL freshly prepared EDC solution and 100 μL 150 mg/mL sulfo-NHS in MES buffer (25 mM, pH 6.0) at RT for 30 min. Excess EDC and sulfo-NHS were separated by two rounds of centrifugation (14,500 × $g$) and resuspension in RO water using 30K MWCO filters (Millipore, UFC503024). The purified beads with activated carboxyl groups were then exposed to 200 μL 35 mM NH2-PEG-COOH (Laysan Bio Inc, NC1641410) and 200 μL 320 mM NH2-PEG-OH (Laysan Bio Inc, NC1641409) in DPBS buffer (pH 8.1) for 3 h at RT to yield a well-shielded PEG layer. Excess PEG molecules were removed by three rounds of centrifugation and resuspension in RO water using 100K MWCO filters (Millipore, UFC510024). For antibody conjugation, the carboxyl groups on PEGylated beads were then activated with 100 μL 100 mg/mL

freshly prepared EDC solution and 100 μL 150 mg/mL sulfo-NHS in MES buffer (25 mM, pH 6.0) at RT for 30 min. After two rounds of centrifugation, the activated beads were then mixed with antibodies at a bead: antibody molar ratio of 1:30 and react for 3 h in HEPES buffer (10 mM, pH 8.3) at RT. The optimal reaction concentration of antibody is 0.5 mg/mL. The bead-Ab conjugates were separated from free antibodies by centrifugation for 3 rounds at 14,500 × $g$. Then the conjugates were resuspended in DPBS buffer (Gibco, 14190136) for use. For aptamer conjugation, the activated beads were mixed with aptamer at a bead: aptamer molar ratio of 1:100 and react for 3 h in HEPES buffer (10 mM, pH 8.3) at RT. Free aptamers were removed by four rounds of centrifugation with 100K MWCO filters and resuspension in DPBS buffer for use (see Supplementary Table 2 for antibody catalog numbers and aptamer sequences).

**pH stability of Rdots**. To assess the effect of pH conditions on Rdots stability, the Rdots were exposed to opti-MEM (Gibco, 31985062) with pH 7.0, 5.5, and 4.5. The pH of opti-MEM was adjusted by adding acid dropwise with constant stirring. The time-dependent Raman intensity was measured through our home-built Raman microscope.

**Metabolic probe preparation**. To prepare $^{13}$C-AA DMEM, 4 mg/mL algae $^{13}$C-amino acid mix (CLM-1548, Cambridge isotope) was dissolved in RO water supplemented with 10% FBS, 1% P/S and other components including vitamin, inorganic salts, and glucose according to DMEM media formula (Invitrogen, 11965). 17-ODYA (Tocris Bioscience, 06-171-0) was dissolved in DMSO and a working stock solution of 4 mM was prepared by 1:6 complexing to BSA (Sigma-Aldrich, A6003). $^{13}$C-EdU was synthesized as the previous reported[53]. AltQ2 (a generous gift from Professor Mikiko Sodeoka) was dissolved in DMSO to get a stock solution of 10 mM for use.

**Cell proliferation assay**. To evaluate the cytotoxicity of Rdots on cells, SKBR3 cells were incubated with 1 nM endocytosis beads for 6 h and then mixed with 10 nM Rdots for surface protein labeling. Cell viability was studied using Live/Dead cell double staining by incubating with 2 μM calcein-AM (Invitrogen, C3099) and 2.5 μM propidium iodide (PI, Sigma-Aldrich, P4864) for 30 min at 37 °C. Fluorescent images were acquired by Olympus confocal microscopy prior to viable/dead cell counting.

**Live-cell surface protein staining**. Cells were dissociated using trypsin-EDTA (Gibco, 25-200-056) on reaching 75% confluence and then harvested in the tube. After two rounds of washing with ice-cold DPBS, cells were resuspended on DPBS buffer with 5 mM MgCl$_2$, 1 mg/mL yeast tRNA (Invitrogen, AM7119), and 1% BSA (Sigma-Aldrich, 05470) to reach 10$^7$ cells/mL. For aptamer annealing, aptamers conjugated Rdots were incubated on a heat block at 90 °C for 4 min and then slowly cool to RT. Then cells were stained with seven-colored Rdot conjugates at a concentration of 10 nM for 30 min on ice. Followed by three rounds of washing with DBS buffer with 5 mM MgCl$_2$ and 1% BSA, cells were attached to a poly-L-lysine coated coverslip (Neuvitro GG12PDL) and mounted onto the microscope for Raman measurement.

**Automated Raman spectrometer**. The schematic of the home-built Raman microscope is shown in Fig. 1. Here, the ×10 objective was underfilled to reach an illumination diameter of ~8 μm. A motorized stage was installed to automatically park on cells identified through bright field. The entire system was controlled through a LabVIEW-based software module (National Instrument).

**Stimulated Raman scattering (SRS) microscopy**. The setup of SRS microscopy has been described previously[26]. Briefly, an integrated laser (Applied Physics and Electronics, Inc., picoEMERALD) was coupled into an inverted laser scanning confocal microscope (Olympus, FV1200). The Stokes beam (1064 nm, 6 ps pulse width) was intensity-modulated at 8 MHz by electro-optic-modulator, and a tunable pump beam (720–990 nm, 5–6 ps pulse width) was produced by the optical parametric oscillator. The laser beams were focused on the sample through a ×25 water immersion objective (Olympus, XLPlan N, 1.05 NA MP). For cellular imaging, 100 mW pump and 400 mW Stokes power were used, with 40 μs time constant and the matching pixel dwell time.

**Multiparameter live-cell profiling**. SKBR3 cells were seeded onto 18 mm round quartz coverslips (Electron Microscopy Sciences, 103302-258) and then maintained in a culture environment for 48 h to reach 90% confluence. Then the culture medium was replaced with $^{13}$C-AA DMEM containing 200 μM 17-ODYA and 50 nM $^{13}$C-EdU for 60 h. For the drug testing samples, cells were subject to the drug treatment simultaneously (see Supplementary Table 3 for specific drug concentration). Followed by three rounds of gentle washing of DPBS buffer, cells were subsequently incubated with three-colored endocytic Rdots at 1 nM in serum-free DMEM medium. Six hours later, cell surface markers were stained with seven-colored Rdots at a concentration of 10 nM for 1 h on ice. After that, cells were washed extensively with DPBS buffer with 5 mM MgCl$_2$ and 1% BSA. Then cells were rinsed with DPBS buffer with 5 mM MgCl$_2$ and 40 μM AltQ2 before mounting onto the microscope for Raman measurement.

**Spectral unmixing processing**. Before the unmixing processing, the acquired spectra from each sample exhibiting a low Pearson correlation value with others (smaller than 0.95) were defined as outlier data and excluded from the analysis. To decipher the contribution of individual Raman probes, spectra unmixing was performed for the amide region and cell-silent region, respectively. The normalized spectra of 16 components (including 14-plexed Raman probes, C=C vibration of PS bead, and $^{12}$C amide peak) in Fig. 5e were employed as reference spectra and characterized by library $\mathbf{M_{amide}}$ and $\mathbf{M_{silent}}$, respectively. After background removal, the trimmed single-cell Raman spectrum $\mathbf{I_{amide}}$ (1500–1700 cm$^{-1}$) and $\mathbf{I_{silent}}$ (2000–2300 cm$^{-1}$) can be deconvolved into the weighted ($\mathbf{\Theta}$) sum of reference spectrum and noise $\mathbf{N}$. The process of linear unmixing can be described as follows:

$$\mathbf{I_{silent}} = \mathbf{M_{silent}} \times \Theta_{silent} + \mathbf{N} \tag{1}$$

$$\mathbf{I_{amide}} = \mathbf{M_{amide}} \times \Theta_{amide} + \mathbf{N} \tag{2}$$

Approximating that the value of $\mathbf{N}$ is negligible for the low-noise spectrum, so here we made an estimation:

$$\mathbf{I_{silent}} \approx \mathbf{M_{silent}} \times \Theta_{silent} \tag{3}$$

$$\mathbf{I_{amide}} \approx \mathbf{M_{amide}} \times \Theta_{amide} \tag{4}$$

As a result,

$$\mathbf{M_{silent}}^{-1} \times \mathbf{I_{silent}} \approx \mathbf{M_{silent}}^{-1} \times \mathbf{M_{silent}} \times \Theta_{silent} \tag{5}$$

which means the decomposed contribution of each component

$$\Theta_{silent} \approx \mathbf{M_{silent}}^{-1} \times \mathbf{I_{silent}} \tag{6}$$

Similarly,

$$\Theta_{amide} \approx \mathbf{M_{amide}}^{-1} \times \mathbf{I_{amide}} \tag{7}$$

The matrix computation was carried out with MATLAB. To verify the linear unmixing model is satisfactory for multiparameter Raman spectral decomposition and robust to measurement noise, the single-cell Raman spectrum acquired was reconstructed with the calculated $\mathbf{\Theta}$. The Pearson correlation coefficient was employed to represent the similarity between reconstructed and ground spectrum and assess the robustness of the unmixing model.

**Reporting summary**. Further information on research design is available in the Nature Research Reporting Summary linked to this article.

## Data availability
All the data supporting this study are available in the article and its Supplementary materials. Source data are provided with this paper.

## Code availability
The MATLAB software package specr for spectral processing can be found at Github[88]. The LabVIEW package RaSpectrometer for the instrument control is not publicly available due to the large file size, but is available from the corresponding author upon reasonable request.

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

## Acknowledgements

We thank Professor Mikiko Sodeoka for her generous gift of AltQ2. W.M. acknowledges grant support of R01 (GM128214) from the National Institute of Health.

## Author contributions
C.C., Z.Z., and W.M. designed the project. C.C. performed the study, analyzed the data, and wrote the paper. Z.Z. designed the optical setup and software for data acquisition automation. N.Q. and S.W. contributed to compounds synthesis. F.H. and W.M. gave technical support and conceptual advice. All authors discussed the results and commented on the manuscript at all stages.

## Competing interests
The authors declare no competing interests.
