## [Peer Review File · Nature Communications]

REVIEWER COMMENTS

Reviewer #1 (Remarks to the Author):

The manuscript reports the demonstration of multi-parameter single-cell profiling based on Raman microspectroscopy using Raman tags. By using the recently reported Rdots technique, in which Raman tags are incorporated in polymer microbeads via swelling-shrinking technology for enhancing the signal intensity, the authors demonstrated 14-plexed live-cell profiling under drug treatments. By virtue of multi-parameter measurements, high-content analyses such as correlation analysis and clustering became possible for live cells. Overall, the experiments were carefully designed so that the presented data support their claims. Although there are some minor issues that need to be addressed before publication, basically I recommend acceptance for publication of this work in Nature Communications.

Minor issues

1. In this work, the authors used a spontaneous Raman micro-spectroscopic setup while they have reported numerous Raman-tag-based imaging based on stimulated Raman scattering microscopy so far. I recommend that they explicitly describe why they used spontaneous Raman spectroscopy in this work because many researchers in the Raman community are aware that the authors' group mainly employs stimulated Raman scattering imaging. Also, I suggest that the authors provide a summary of the pros and cons of spontaneous and stimulated Raman scattering measurements as it may be helpful for broad readers of Nature Communications.

2. One great point of spontaneous Raman spectroscopy compared to the coherent Raman techniques (and even to fluorescence, which sometimes lacks concentration linearity) is its strict linearity in terms of concentration dependence. I believe, in their demonstrations, it is possible to quantify how many Rdots were incorporated in each cell by calibrating the measured Raman intensities by those of pure Rdots solutions. Such information of the absolute numbers of proteins or polymer beads incorporated by endocytosis would be a valuable piece of information and may be biologically significant.

3. In Table 1, Carbow 2098 should probably be Carbow 2092. If not, the information about Carbow 2092 is missing and should be provided.

4. In Fig. 3e-3g, the intensities are on the same order although Raman signal intensities from three Rdots are significantly different in Fig. 1e. Does this mean the incorporation of microbeads is totally different for those three beads or did they normalize the intensities?

5. They used relatively high power (400 mW) for their measurements. Photo-induced damage to the cell should be discussed if they claim this technique is compatible with live-cell profiling. For example, I suggest the authors evaluate cell viability after laser irradiation.

6. On page 15, they describe that spectral unmixing was performed through the least-squares method. The basis set they used and fitting error for each parameter for the analysis should be clarified.

7. On page 7, the characters after "amide" are corrupted. Is this a PDF conversion error or present in the Word version too? Anyway, the authors should correct it.

Reviewer #2 (Remarks to the Author):

This manuscript reports the use of polymer-encapsulated Raman-active molecules for multiplexed bioimaging. Even though the authors show high quality analysis of imaging with extended multiplex capability, my opinion is that the conceptual novelty is hindered by the previous publication of ref. 32. The concept appears exactly the same to me, some figures are even identical, and the main difference is the increased number of labels, so the question arises: given that the previous manuscript has very recently been accepted (not even published online), why

not including the extended library in the same manuscript?

Therefore, I would recommend that the newer manuscript is expanded to in vivo imaging or practical applications of flow cytometry, and submitted to a more specialised journal.

Reviewer #3 (Remarks to the Author):

In their article "Super-multiplexed live-cell profiling by Raman microscopy", the authors present the use of their Raman probes in the case of live cell phenotyping through the observation of membrane receptors.

The article present impressive results that are supported by the extensive results and validation data. They are novel and of interest to the community. There are however some small points which should be clarified in my opinion, and I partly disagree with some of the authors claims, as detailed below.

First, as a general comment, I think that the title and abstract are not necessarily representative of the core results of the paper. The fact that it is employing Raman tags is not present in the title, and only very shortly mentioned in the abstract. It might be beneficial to make this more explicit. Secondly, the authors employ two measurement approaches: spontaneous Raman for all main results, and SRS for some imaging validation. This is mentioned in the results, but being more explicit about this that all main results are obtained by resonant spontaneous Raman in the introduction could help clarity too.

My main issue about the authors' claim is about the comparison with fluorescence. While they indeed demonstrate impressive 14-channel multiplexing, they are claiming that their approach relies on a simpler system (true) and can perform better due to complex spectral compensation procedures in fluorescence. However, there is also significant spectral overlap between their probes (Fig. 1d and 4b), requiring unmixing procedures. Several points derive from this:

- For example, 14-channel fluorescence is indeed challenging in fluorescence, but possible especially considering new development in UV and near-IR dyes. While fluorescence has indeed large emission regions, the overall available spectral range is much larger too (400-800 nm)
- I am not sure to follow the unmixing procedure. I understand that this derives from an earlier publication, but more details may be warranted, and it was previously demonstrated with far less channels and spectral overlap.
- Am I correct in understanding that the linear decomposition is performed on the signal of the average population (Fig. 4d-e), and then applied back to single-cell data (Fig. 4f-g)? If yes, I find surprising that the correlation is so good, considering that cell contributions and staining efficiency should vary cell to cell. Is the correlation comparable on all cells?
- Multiplexed fluorescence indeed requires complex spectral calibration, such as by measuring all channels independently to build a correlation matrix, but this also ensures more independence between channels, which is not demonstrated by the authors. This implies that some of the correlations they observe in further experiments might be due to such interdependence.

I am surely not necessarily suggesting additional experiments, especially considering the extensive work the authors provide already, but these points weaken some of the claims given by the authors, and should be reflected in the comments and discussion.

As a related side note, I imagine that the widening of the beam used to measure whole-cell spectra should also be widening peaks as the pinhole is also larger. How does that influence the spectral overlap between channels?

In addition to that major point, there are several minor points to consider:

- Figure 2c, possibly also 3b-d: If available, the addition to the brightfield might help the understanding of the SRS image. The overall cell structure is hard to delineate.
- Figure 3e-g: sample size is not indicated, significance should be given.
- I am not sure to understand the biological significance of employing different unrelated cell lines

(also from different species) in the results of Fig. 3j-m. While it shows the ability of the probes to identify the difference in endocytosis (as already demonstrated in previous results in the manuscript), having large differences is hardly surprising.

- Sample size is missing in Fig. S5
- Fig. S6 indeed shows a link between the signal and the presence of the receptor under study, but not correlation with the expression level. This would require further tests to show the linearity of increase with expression levels. This also relates with independence of the channels, as mentioned above. Are the curves in Fig. S6 from single cells or averaged?
- Fig. S7 is missing sample size, and statistical significance (or absence of it) should be indicated.
- Page 7: some characters have encoding problems, I suppose it is supposed to be "Amide I" and "Amide III"
- Page 9: I agree that single-cell measurements is far more informative than average populations, but as this technique mostly compares with fluorescence and mass cytometry, which techniques are the authors referring to?
- Page 11: The authors claim that it would be straightforward to extend that measurement approach to CARS and SRS to improve throughput, but on the other hand, due to the very small excitation volume in nonlinear optics, whole-cell spectra would have to be recorded in a completely different manner, which could very well reduce throughput as well. A more detailed discussion would be required to make such a claim.

RESPONSE TO REVIEWER COMMENTS

Reviewer #1 (Remarks to the Author):

The manuscript reports the demonstration of multi-parameter single-cell profiling based on Raman microspectroscopy using Raman tags. By using the recently reported Rdots technique, in which Raman tags are incorporated in polymer microbeads via swelling-shrinking technology for enhancing the signal intensity, the authors demonstrated 14-plexed live-cell profiling under drug treatments. By virtue of multi-parameter measurements, high-content analyses such as correlation analysis and clustering became possible for live cells. Overall, the experiments were carefully designed so that the presented data support their claims. Although there are some minor issues that need to be addressed before publication, basically I recommend acceptance for publication of this work in Nature Communications.

Response:

We thank the reviewer's appreciation and highly supportive comments on our paper. These constructive remarks have enabled us to further improve our manuscript. Below, please find a point-by-point response. We hope these can help clarify the reviewer's remaining concerns.

Minor issues

1. In this work, the authors used a spontaneous Raman micro-spectroscopic setup while they have reported numerous Raman-tag-based imaging based on stimulated Raman scattering microscopy so far. I recommend that they explicitly describe why they used spontaneous Raman spectroscopy in this work because many researchers in the Raman community are aware that the authors' group mainly employs stimulated Raman scattering imaging. Also, I suggest that the authors provide a summary of the pros and cons of spontaneous and stimulated Raman scattering measurements as it may be helpful for broad readers of Nature Communications.

Response:

Indeed, the previous work from our group was mostly achieved with stimulated Raman scattering (SRS) microscopy. As a powerful vibrational imaging technique, SRS offers tremendous signal amplification and can map the distribution of chemical bonds in three-dimensional space and real-time. However, in the current paper, the spontaneous Raman micro-spectroscopy was applied here, as spontaneous Raman setup is much more accessible and cost-effective (see the newly added Scheme 1 in the revised manuscript) for most biomedical researchers when compared to the SRS microscope system. We believe that our demonstration of the simple spontaneous Raman setup will better promote the wide utility of our newly developed probe set.

We agree with the reviewer that SRS microscopy will be an attractive option for single-cell profiling especially with even higher throughput. In fact, we plan to publish SRS-based multiplexed single-cell profiling in our follow-up work. One technical issue is how to acquire and quantify the single cell signal as the laser focus of SRS excitation is much smaller than the cellular volume. This will be addressed in our future work that employs SRS microscopy.

As suggested by the reviewer, the cost-effectiveness of spontaneous Raman micro-spectroscopy has been emphasized in our revision, and the future direction of employing SRS for even higher throughput has also been better elaborated in the final discussion part of the revision.

2. One great point of spontaneous Raman spectroscopy compared to the coherent Raman techniques (and even to fluorescence, which sometimes lacks concentration linearity) is its strict linearity in terms of concentration dependence. I believe, in their demonstrations, it is possible to quantify how many Rdots were incorporated in each cell by calibrating the measured Raman intensities by those of pure Rdots solutions. Such information of the absolute numbers of proteins or polymer beads incorporated by endocytosis would be a valuable piece of information and may be biologically significant.

Response:

We thank the reviewer's suggestion, which is rather insightful. To quantify the number of incorporated Rdots, we firstly need to measure the absolute Raman intensity of a single Rdot. The linear relationship between Raman intensity at 3055cm^{-1} (C–H vibration in styrene) and the volume of a single bead (Figure R1) implies that the density of polystyrene (PS) beads at different sizes is identical. So we can take the $3\ \mu\text{m}$ PS beads to estimate the Raman intensity of a single Rdot. The volume size of a single $3\ \mu\text{m}$ bead equals to that of 4.2×10^5 Rdots at $40\ \text{nm}$. Since the Raman intensity of $3\ \mu\text{m}$ PS bead at $3055\ \text{cm}^{-1}$ is 7.8×10^3 counts/200 mW*4s (Figure R2), the intensity of $3055\ \text{cm}^{-1}$ peak for single $40\ \text{nm}$ Rdot is $7.8 \times 10^3 / 4.2 \times 10^5 = 1.9 \times 10^{-2}$ counts/200 mW*4s, or 9.3×10^{-3} counts/400 mW*s.

Figure R1. The linear relationship between Raman intensity of 3055cm^{-1} and volume of a single bead.

Figure R2. (a) Bright field image of suspended $3\ \mu\text{m}$ polystyrene beads, (b) Raman spectrum of single $3\ \mu\text{m}$ PS bead (indicated by the arrow in a). Excitation: $532\ \text{nm}$ (200 mW), acquisition time: 4 s.

For Rdot2218, the Raman signal ratio of 2218 cm^{-1} peak (C≡C vibration in Carbow2218) and 3055 cm^{-1} peak is calculated: $R_{2218/3055} = 6.0$ (Figure R3). So the intensity of 2218 cm^{-1} peak for single 40 nm Rdot is $9.3 \times 10^{-3} \times 6.0 = 5.6 \times 10^{-2}$ counts/400 mW*s.

As shown in Figure 2b in the manuscript, the absolute Raman intensity at 2218 cm^{-1} of Rdot2218-CD44 on a single HeLa cell is 500 counts/400 mW*s. As a result, the number of Rdot2218-CD44 on a single cell is $500/5.6 \times 10^{-2} = 9.0 \times 10^3$.

As reported, the abundance of CD44 protein in HeLa cells is about 2 million copies/cell¹. Then we could estimate that the labeling efficiency of Rdot2218-CD44 on HeLa cell is about 4.5%. Here, we provide a general approach to estimate the labeling efficiency of Rdots. This value greatly depends on the copy number and membrane assemblies of proteins and thus varies for different proteins and cell types.

Figure R3. Raman spectrum of 40 nm Rdot2218 solution. Excitation: 532 nm (400 mW), acquisition time: 0.5 s.

A similar calculation can be performed to quantify the number of internalized endocytic Rdots. Taking Rdot2092-120nm as an example, the volume size of a single 3 μm PS bead equals to that of 1.6×10^4 Rdots at 120 nm diameter. As a result, the intensity of the 3055 cm^{-1} peak for a single 120 nm Rdot is 0.25 counts/400 mW*s. The Raman signal ratio of 2092 cm^{-1} peak and 3055 cm^{-1} peak is calculated: $R_{2092/3055} = 6.0$ (Figure R4). So the intensity of 2092 cm^{-1} peak for a single 120 nm Rdot is 1.50 counts/400 mW*s. As shown in Figure 3a in the manuscript, the absolute Raman intensity of 2092 cm^{-1} peak acquired after incubated with endocytic Rdots for 6 hours is about 150 counts/400 mW*s, indicating the number of internalized Rdot2092 is $150/1.50=100$. Under a similar calculation, the numbers of internalized Rdot2118-70nm and Rdot2207-40nm were 1300 and 9000 respectively.

Figure R4. Raman spectrum of 120 nm Rdot2092 solution. Excitation: 532 nm (400 mW), acquisition time: 1 s.

We have added the above discussion in the SI and hope these help showcase the quantitative capability of spontaneous Raman and strengthen the biological significance of our observation.

3. In Table 1, Carbow 2098 should probably be Carbow 2092. If not, the information about Carbow 2092 is missing and should be provided.

Response:

Carbow 2098 in Table 1 should be Carbow 2092. We appreciate the review's careful inspection, and have corrected the typos.

4. In Fig. 3e-3g, the intensities are on the same order although Raman signal intensities from three Rdots are significantly different in Fig. 1e. Does this mean the incorporation of microbeads is totally different for those three beads or did they normalize the intensities?

Response:

In Figure 3e-g, the Raman intensity of single-type Rdots was normalized by the mean value of the control sample respectively. That's why the intensities are on the same order. Nonetheless, we totally agree that the cellular incorporation of three Rdots is not identical and should be size-dependent. As calculated in our response to Comment 2 above, the numbers of internalized 40 nm, 70 nm and 120 nm Rdots were 9000, 1300 and 100, respectively.

5. They used relatively high power (400 mW) for their measurements. Photo-induced damage to the cell should be discussed if they claim this technique is compatible with live-cell profiling. For example, I suggest the authors evaluate cell viability after laser irradiation.

Response:

400 mW might seem a relatively high power for live-cell measurements. But we would like to remind the reviewer that we have made one key modification to our spontaneous Raman instrument: the laser spot is significantly expanded from the conventional submicron size to ~8 μm to cover the entire cell (see Scheme 1 in the revised manuscript). As a result of the beam expansion, our power density is about 10 times lower than conventional Raman microspectroscopy. Indeed, no obvious photo-damage was observed during measurements with such a low power density.

6. On page 15, they describe that spectral unmixing was performed through the least-squares method. The basis set they used and fitting error for each parameter for the analysis should be clarified.

Response:

We appreciate the reviewer's advice on spectral unmixing. To make this part more self-contained, we've added additional details about unmixing algorithm in the Methods section on Page 16.

Spectral unmixing processing. To decipher the contribution of individual Raman probes, spectra unmixing was performed for the amide region and cell silent region, respectively. The normalized spectra of 16 components (including 14-plexed Raman probes, C=C vibration of PS bead and ^{12}C amide peak) in Figure 4e were employed as reference spectra and characterized by library $\mathbf{M}_{\text{amide}}$ and $\mathbf{M}_{\text{silent}}$, respectively. After background removal, the trimmed single-cell Raman

spectrum \mathbf{I}_{amide} ($1500\text{ cm}^{-1} - 1700\text{ cm}^{-1}$) and \mathbf{I}_{silent} ($2000\text{ cm}^{-1} - 2300\text{ cm}^{-1}$) can be deconvolved into the weighted (Θ) sum of reference spectrum and noise \mathbf{N} . The process of linear unmixing can be described as follows:

$$\mathbf{I}_{silent} = \mathbf{M}_{silent} \times \Theta_{silent} + \mathbf{N};$$

$$\mathbf{I}_{amide} = \mathbf{M}_{amide} \times \Theta_{amide} + \mathbf{N};$$

Approximating that the value of \mathbf{N} is negligible for low-noise spectrum, here we made an estimation:

$$\mathbf{I}_{silent} \approx \mathbf{M}_{silent} \times \Theta_{silent};$$

$$\mathbf{I}_{amide} \approx \mathbf{M}_{amide} \times \Theta_{amide};$$

As a result, $\mathbf{M}_{silent}^{-1} \times \mathbf{I}_{silent} \approx \mathbf{M}_{silent}^{-1} \times \mathbf{M}_{silent} \times \Theta_{silent}$, which means the decomposed contribution of each component $\Theta_{silent} \approx \mathbf{M}_{silent}^{-1} \times \mathbf{I}_{silent}$. Similarly, $\Theta_{amide} \approx \mathbf{M}_{amide}^{-1} \times \mathbf{I}_{amide}$. The matrix computation was carried out with MATLAB. To verify the linear unmixing model is satisfactory for multiparameter Raman spectral decomposition and robust to measurement noise, the single-cell Raman spectrum acquired was reconstructed with the calculated Θ . The Pearson correlation coefficient was employed to represent the similarity between reconstructed and ground spectrum and assess the robustness of the unmixing model.

Additionally, the fitting error (Pearson correlation coefficients distribution between reconstructed and raw single-cell spectra) was plotted as Figure S14. The high correlation value (larger than 0.988 mostly) validates the robustness of the unmixing model.

Figure S14. The Pearson correlation coefficients distribution between reconstructed and raw spectra. The high correlation value (larger than 0.988 mostly) validates the robustness of the unmixing model. $N_{Control}=213$; $N_{CytoD}=189$; $N_{CHX}=167$; $N_{trastuzumab}=235$; $N_{Cisplatin}=136$; $N_{H2O2}=267$.

Hope these demonstrations will make the unmixing procedure clear.

7. On page 7, the characters after “amide” are corrupted. Is this a PDF conversion error or present in the Word version too? Anyway, the authors should correct it.

Response:

The Roman numbers were corrupted during PDF conversion, it should be "Amide I" and "Amide II". We have corrected them in the revision.

Reviewer #2 (Remarks to the Author):

This manuscript reports the use of polymer-encapsulated Raman-active molecules for multiplexed bioimaging. Even though the authors show high quality analysis of imaging with extended multiplex capability, my opinion is that the conceptual novelty is hindered by the previous publication of ref. 32. The concept appears exactly the same to me, some figures are even identical, and the main difference is the increased number of labels, so the question arises: given that the previous manuscript has very recently been accepted (not even published online), why not including the extended library in the same manuscript?

Therefore, I would recommend that the newer manuscript is expanded to in vivo imaging or practical applications of flow cytometry, and submitted to a more specialised journal.

Response:

We appreciate the reviewer's comments. It appears that the reviewer thinks our paper is an incremental improvement of the first paper (ref. 32), which should be included in the same paper. However, we respectfully disagree with the reviewer's critical assessment that "*the main difference is the increased number of labels*". As detailed below, the reviewer might have overlooked the following four major differences between the current work and the previous ref. 32.

(1) The scope of the current paper is different from the previous ref. 32. Ref. 32 is mainly focusing on the development of the Rdots as a novel Raman probe, as most of the content was about the design and characterization of the versatility and brightness of Rdots. The immunohistology application in that paper was only to demonstrate the potential of Rdots. However, in the current paper, we extensively harnessed Rdots to profile surface markers and endocytosis, and most of the content was about developing and demonstrating a high throughput live-cell profiling platform (but not the Rdots themselves) to make biological discoveries. It's a common practice in the field to put such diverse contents into separate papers.

(2) More importantly, the previous ref. 32 and the current manuscript address totally distinctive biological problems. Ref. 32 tackles immunohistology to analyze the spatial pattern of protein distribution, and the final data is the spatial-resolved images of specific markers inside cells and tissues. In contrast, in the new manuscript, we aim to address the challenge of single-cell phenotyping in a multiplexed and high-throughput manner to analyze cell heterogeneity. The final delivery is a set of quantitative biological parameters of individual cells. Important to note that the spatial information was not measured in single-cell phenotyping. Instead, what really matters is the correlation between multiple parameters at the single-cell level and how cells behave differently in a large population. To achieve this goal, our new manuscript has pioneered the integrative profiling of cell surface proteins, endocytosis activities, and metabolic dynamics to realize the highest Raman-based multiplexing reported to date. The applications of endocytosis activities and metabolic dynamics were not presented in ref.32 at all. With such different biological applications, it's practically impossible to combine the two papers together into a single article.

(3) The current paper employed instrumentation that is totally different from that used in ref.32. In Ref.32, we used stimulated Raman scattering (SRS) microscopy to image the spatial distribution of protein targets inside cells and tissues, as SRS microscopy exhibits advantage in spatial resolution and imaging speed. In contrast, in this paper, we used a home-modified spontaneous Raman microscope to profile surface markers, endocytosis, and metabolic activities, as spatial

information is no longer crucial in single-cell phenotyping. Compared to SRS microscope which is rather expensive and narrow-banded, the spontaneous Raman microscope is significantly cheaper, more widely available and more straightforward and robust in acquiring multiplex Raman spectra. Yet, without the amplification from SRS, it is not clear from the onset whether spontaneous Raman spectroscopy will generate enough signal for surface protein markers. To improve the signal to noise ratio, we modified our spontaneous Raman microscope to excite and collect the signal from a whole cell instead of using a commercial confocal Raman spectrometer that only reads signals from a diffraction-limited volume. Such a whole-cell Raman microspectrometer allows us to achieve SNR boost by more than 100 times while reducing sampling errors. We have now added a new Scheme 1 in the revision to emphasize this key point of physical instrumentation.

(4) Finally, we also made efforts to tailor probe functionalization for the task of high-throughput multiplex profiling. For example, the modality of Rdots in this manuscript is markedly different from ref.32. Color-wise, we expanded the previously reported 6-color Rdots palette to 10 colors by optimizing the encapsulation of more Raman probes, including some probes with negative charges, which was not used in ref.32. Size-wise, 20 nm Rdots were employed in ref. 32 for immunohistology. The compact size enables Rdots to diffuse into tissues and stain intracellular markers. In contrast, the compact size of Rdots is not the major concern in the current paper, as most surface proteins locate on the cell membrane. As a result, instead of using the same 20nm Rdots, we chose 40 nm Rdots and demonstrated such size increase will lead to significant brightness increase, which is crucial to successful profiling with spontaneous Raman microscopy. Surface-wise, surface functionalization of Rdots has also been optimized for the new applications. A longer PEG chain was applied to minimize the non-specific binding. Besides, as a cost-effective alternative to antibodies, aptamers were firstly incorporated on Rdots for live-cell profiling, which was not shown in ref.32.

To summarize, the focuses of the two papers are distinct; the biological problems that have been addressed in the two papers is distinct; the physical instrumentation employed in the two articles are distinct; and substantial probe optimization and functionalization have been tailored for the task of single-cell profiling. Therefore we believe that this set of differences warrant enough novelty of the current paper over the previous one.

We thank the review's insightful suggestion of performing flow cytometry application. We're actually working along this direction. However, we wish to demonstrate the spectroscopy feasibility of using Rdots for single-cell multiplex profiling as a pioneering paper first. As we discussed above, it is not clear from the onset whether spontaneous Raman spectroscopy will generate enough signal to noise ratio (SNR) for surface protein markers. Indeed, we need to develop our whole-cell Raman micro-spectrometer to boost SNR boost by 100 times. Hence, the feasibility demonstration of the current paper, which is highly non-trivial, would serve as the foundation for future engineering work to build on. The realization of practical flow cytometry will require sophisticated hardware and very much beyond the scope of the current paper. In fact, the realization of flow cytometry with Raman techniques will deserve a piece of publication by itself (see a recent example by Yuta Suzuki et. al. *PNAS*, 2019).

We hope these discussions could help convince the reviewer. We have also further emphasized these specific points in the revised version of our manuscript.

Reviewer #3 (Remarks to the Author):

In their article "Super-multiplexed live-cell profiling by Raman microscopy", the authors present the use of their Raman probes in the case of live cell phenotyping through the observation of membrane receptors.

The article presents impressive results that are supported by the extensive results and validation data. They are novel and of interest to the community. There are however some small points which should be clarified in my opinion, and I partly disagree with some of the authors claims, as detailed below.

Response:

We greatly appreciate the reviewer's highly supportive comments and many valuable suggestions. Below, please find a point-by-point response. We hope these can help clarify the reviewer's remaining concerns.

1. First, as a general comment, I think that the title and abstract are not necessarily representative of the core results of the paper. The fact that it is employing Raman tags is not present in the title, and only very shortly mentioned in the abstract. It might be beneficial to make this more explicit.

Response:

We've modified the title and abstract to highlight the development of Raman tags.

Modified title: Super-Multiplexed Live-Cell Profiling with Raman probes

Sentences in abstract highlighting the development of Raman tags: Here, we devised a novel super-multiplexed Raman probe panel with sharp and mutually resolvable Raman peaks to simultaneously quantify cell surface proteins, endocytosis activities, and metabolic dynamics of individual live cell.

2. Secondly, the authors employ two measurement approaches: spontaneous Raman for all main results, and SRS for some imaging validation. This is mentioned in the results, but being more explicit about this that all main results are obtained by resonant spontaneous Raman in the introduction could help clarity too.

Response:

We thank the reviewer's kind suggestion, and have explicitly stated that all results were obtained by spontaneous Raman in the Introduction section. Besides, we've added Scheme 1 in the revised manuscript to make the workflow and employed system clear.

3. My main issue about the authors' claim is about the comparison with fluorescence. While they indeed demonstrate impressive 14-channel multiplexing, they are claiming that their approach relies on a simpler system (true) and can perform better due to complex spectral compensation procedures in fluorescence. However, there is also significant spectral overlap between their probes (Fig. 1d and 4b), requiring unmixing procedures. Several points derive from this:
- For example, 14-channel fluorescence is indeed challenging in fluorescence, but possible especially considering new development in UV and near-IR dyes. While fluorescence has indeed large emission regions, the overall available spectral range is much larger too (400-800 nm).

Response:

We looked through many fluorescence papers and found >10 plexed fluorescence spectral imaging is seldomly reported or utilized in the literature. For example, Figure R5 shows the emission spectra detected in six-colored fluorophore labeled cells. Unlike the reference spectra (Figure R5b), the cell readouts exhibit a much lower sampling rate (Figure R5c), which will greatly increase the uncertainty during spectral unmixing and thus reduce the quantitative accuracy. This may hamper the development of multiplexed fluorescence spectral imaging.

Nevertheless, we agree with the reviewer that it is theoretically possible for fluorescence microscopy to improve the multiplexing ability with a broadened spectral range going from UV to near IR. In fact, we have explicitly stated in our original manuscript that the implementation of 17-color fluorescent flow cytometry² was reported before and that super-multiplexed fluorescence spectral imaging is achievable with an elaborate dye arrangement.

Figure R5². (a) Optical setup for fluorescence spectral imaging; (b) Reference spectra of six fluorophores; (c) The emission of the indicated fluorophores in images of singly labeled cells.

Hence, instead of claiming that our method is definitely better than fluorescence, we modified our claim in our revision to say that Raman and fluorescence are two complementary strategies for super-multiplexed measurements with their own pros and cons. In particular, our Raman-based multiplexing platform provides a more straightforward, reliable and cost-effective choice, which will be more affordable to most researchers.

4. - I am not sure to follow the unmixing procedure. I understand that this derives from an earlier publication, but more details may be warranted, and it was previously demonstrated with far less channels and spectral overlap.

Response:

We appreciate the reviewer's advice on spectral unmixing analysis. To make this part more self-contained, we've added additional details in the Methods section on Page 16 as follows:

Spectral unmixing processing. To decipher the contribution of individual Raman probes, spectra unmixing was performed for the amide region and cell silent region, respectively. The normalized spectra of 16 components (including 14-plexed Raman probes, C=C vibration of PS bead and ¹²C amide peak) in Figure 4e were employed as reference spectra and characterized by library M_{amide} and M_{silent} , respectively. After background removal, the trimmed single-cell Raman spectrum I_{amide} (1500 cm^{-1} – 1700 cm^{-1}) and I_{silent} (2000 cm^{-1} – 2300 cm^{-1}) can be deconvolved into the weighted (Θ) sum of reference spectrum and noise N . The process of linear unmixing can be described as follows:

$$I_{silent} = M_{silent} \times \Theta_{silent} + N;$$

$$I_{amide} = M_{amide} \times \Theta_{amide} + N;$$

Approximating that the value of N is negligible for low-noise spectrum, here we made an estimation:

$$\underline{\mathbf{I}}_{\text{silent}} \approx \underline{\mathbf{M}}_{\text{silent}} \times \underline{\mathbf{O}}_{\text{silent}};$$

$$\underline{\mathbf{I}}_{\text{amide}} \approx \underline{\mathbf{M}}_{\text{amide}} \times \underline{\mathbf{O}}_{\text{amide}};$$

As a result, $\underline{\mathbf{M}}_{\text{silent}}^{-1} \times \underline{\mathbf{I}}_{\text{silent}} \approx \underline{\mathbf{M}}_{\text{silent}}^{-1} \times \underline{\mathbf{M}}_{\text{silent}} \times \underline{\mathbf{O}}_{\text{silent}}$, which means the decomposed contribution of each component $\underline{\mathbf{O}}_{\text{silent}} \approx \underline{\mathbf{M}}_{\text{silent}}^{-1} \times \underline{\mathbf{I}}_{\text{silent}}$. Similarly, $\underline{\mathbf{O}}_{\text{amide}} \approx \underline{\mathbf{M}}_{\text{amide}}^{-1} \times \underline{\mathbf{I}}_{\text{amide}}$. The matrix computation was carried out with MATLAB. To verify the linear unmixing model is satisfactory for multiparameter Raman spectral decomposition and robust to measurement noise, the single-cell Raman spectrum acquired was reconstructed with the calculated $\underline{\mathbf{O}}$. The Pearson correlation coefficient was employed to represent the similarity between reconstructed and ground spectrum and assess the robustness of the unmixing model.

Additionally, the fitting error (Pearson correlation coefficients) distribution between the reconstructed and the raw single-cell spectra was plotted as Figure S14. The high correlation value (larger than 0.988 mostly) validates the robustness of our unmixing model.

Figure S14. The Pearson correlation coefficients distribution between reconstructed and raw spectra. The high correlation value (larger than 0.988 mostly) validates the robustness of the unmixing model. $N_{\text{Control}}=213$; $N_{\text{CytoD}}=189$; $N_{\text{CHX}}=167$; $N_{\text{trastuzumab}}=235$; $N_{\text{Cisplatin}}=136$; $N_{\text{H2O2}}=267$.

Hope these demonstrations make the unmixing procedure clear.

5. - Am I correct in understanding that the linear decomposition is performed on the signal of the average population (Fig. 4d-e), and then applied back to single-cell data (Fig. 4f-g)? If yes, I find surprising that the correlation is so good, considering that cell contributions and staining efficiency should vary cell to cell. Is the correlation comparable on all cells?

Response:

We appreciate the reviewer's question regarding the unmixing process. However, the reviewer might have misunderstood our data by thinking that we directly applied the decomposed results of averaged spectra back to the single-cell spectrum. In fact, we performed linear unmixing on averaged population (Figure 4e, updated Figure 4d) first to introduce the principle of intensity decomposition for 14-plexed cell Raman spectra. Right after this, Figure 4f (updated Figure 4e) in our original manuscript showed the implementation of unmixing on the actual single-cell spectrum.

To evaluate the unmixing accuracy, we then reconstructed the single-cell spectrum according to the retrieved intensity matrix acquired in Figure 4f (updated Figure 4e). The high Pearson correlation coefficients between the original and the reconstructed single-cell spectrum in Figure 4g (updated Figure 4f) suggest that the signal-to-noise of the single-cell spectrum is good enough and the nearby cross-talk between channels is acceptable.

We do agree with the reviewer that cell-to-cell variations could be observed for the same batch of samples, considering the inherent cellular heterogeneity. Our experimental observation also supports this consideration (three different single-cell spectra shown in Figure R6).

Figure R6. Multiple single-cell Raman spectrum acquired from the same batch. The spectrum shape of different cells is not exactly the same.

6. - Multiplexed fluorescence indeed requires complex spectral calibration, such as by measuring all channels independently to build a correlation matrix, but this also ensures more independence between channels, which is not demonstrated by the authors. This implies that some of the correlations they observe in further experiments might be due to such interdependence.
Response:

We agree with the reviewer that a reference matrix for spectral calibration ensures more independence between channels. In fact, that's exactly what we did in unmixing. The normalized Raman spectra of 16 components (including 14-plexed Raman probes, C=C vibration of PS bead and ¹²C amide peak) in Figure 4d were measured independently to serve as a reference matrix in our unmixing processing. As a result, the correlations we observed in multiparameter profiling should not be artifacts from interdependent cross-talk.

To clear the potential confusion, we've added some details in the Methods section, as we discussed in response to Comment 4 above. We hope these can help clarify the remaining concerns and misunderstandings of the reviewer.

I am surely not necessarily suggesting additional experiments, especially considering the extensive work the authors provide already, but these points weaken some of the claims given by the authors, and should be reflected in the comments and discussion.

Response:

We thank the reviewer's appreciation and kind consideration. We've added some discussions and modified the manuscript accordingly to clarify some claims.

As a related side note, I imagine that the widening of the beam used to measure whole-cell spectra should also be widening peaks as the pinhole is also larger. How does that influence the spectral overlap between channels?

Response:

We thank the review's question regarding the pinhole size. In our application, the pinhole size was optimized from two aspects. First, as showed in Figure R7, as the pinhole size increases, the peak widths of Carbow dyes were slightly broadened from 14 cm^{-1} ($50\text{ }\mu\text{m}$ pinhole) to 17 cm^{-1} ($300\text{ }\mu\text{m}$ pinhole), which slightly decrease the spectral resolvability and affect the performance of unmixing. Note that the spectral broadening is relatively minor going from 14 to 17 cm^{-1} . Second, the pinhole size should also be optimized to ensure an integrated whole-cell acquisition spatially. A smaller pinhole will block the intensity of single cell, while a larger one will include too much background from substrate and solution. Given all these factors, a $300\text{ }\mu\text{m}$ pinhole was employed for single-cell acquisition. We believe that a slight peak broadening from 14 cm^{-1} to 17 cm^{-1} is acceptable to get a high unmixing accuracy (Figure S14).

Figure R7. Normalized Raman spectra of Carbow2218 (150 mM) solution over multiple detection pinholes (a) $300\text{ }\mu\text{m}$; (b) $100\text{ }\mu\text{m}$ (c) $50\text{ }\mu\text{m}$.

We hope the above evidence could clear the confusion and convince the reviewer.

In addition to that major point, there are several minor points to consider:

- Figure 2c, possibly also 3b-d: If available, the addition to the brightfield might help the understanding of the SRS image. The overall cell structure is hard to delineate.

Response:

We thank the reviewer's helpful suggestion, and have added the bright field image and the SRS image of CH_3 protein channel in Figure S6 and Figure S8 to delineate the cell structure.

Figure S6. Bright-field image (a), SRS image of frequency of 2940 cm^{-1} (b) and 2194 cm^{-1} (c) of Rdot2194-EpCAM positively stained HeLa cells, scale bar: $10\text{ }\mu\text{m}$. Indicated by the bright-field and CH_3 protein channels at 2940 cm^{-1} , most Rdots well located on the cell membrane surface, confirming the specific recognition.

Figure S8. Bright-field image (a), SRS image of cell protein (b), 70 nm Rdots (c) and merged channel (d) of HeLa cells incubated with 70 nm Rdots for 6 hours, scale bar: 10 μm . Indicated by the bright-field and protein channels at 2940 cm^{-1} , most Rdots tend to accumulate in the perinuclear region.

- Figure 3e-g: sample size is not indicated; significance should be given.

Response:

We appreciate the suggestion from the reviewer and have added sample size and significance to Figure 3e-g.

- I am not sure to understand the biological significance of employing different unrelated cell lines (also from different species) in the results of Fig. 3j-m. While it shows the ability of the probes to identify the difference in endocytosis (as already demonstrated in previous results in the manuscript), having large differences is hardly surprising.

Response:

We agree with the reviewer that having large endocytic differences between cell lines is not a totally new discovery in the biological sense. Instead, the key point in Figure 3j-m is mainly for the technical comparison between single-plex and multiplex profiling.

We have noted that significant cell-to-cell variations within a single cell type have caused a wide distribution in the histogram for each Rdot channel (Figure 3j-l). Such single-plex difference is thus overwhelmed by the cell-to-cell variations and has less discriminate power to distinguish three cell lines. Interestingly, the multiple-sized endocytic beads under t-SNE plot significantly improved the separation between different cell lines (Figure 3m), highlighting the greater power of multiplexed endocytic profiling in cell-type identification compared to single-plex analysis. This result lays the foundation for our subsequent 14-plex single-cell profiling.

- Sample size is missing in Fig. S5

Response:

We appreciate the suggestion from the reviewer, and have added the sample size to the Figure S5.

- Fig. S6 indeed shows a link between the signal and the presence of the receptor under study, but not correlation with the expression level. This would require further tests to show the linearity of increase with expression levels. This also relates with independence of the channels, as mentioned above. Are the curves in Fig. S6 from single cells or averaged?

Response:

We appreciate this valuable suggestion the reviewer made. To better demonstrate the linear correlation between the Raman intensity and expression level, an additional discussion has been added in the manuscript.

In Figure S6 (updated Figure S7), we tested seven-colored Rdots conjugates for surface protein measurements. Among them, surface proteins CD44, EpCAM, HER2, EGFR, MUC1, and nucleolin were the cases to show a qualitative link between the signal and the presence of the receptors, because they all exhibit negligible expression on Jurkat cells and positively expressed in HeLa or SKBR3 cells.

More quantitatively, both Jurkat and SKBR3 show moderate expressions of CD55 protein. According to the human protein atlas database (www.proteinatlas.org/ENSG00000196352-CD55/cell), the normalized expression (NX) of CD55 is 1.5 for Jurkat cell and 10.1 for SKBR3 cell. Note that, in Figure S7d, the Raman peak intensity of SKBR3 cells is about 5 times larger than Jurkat cells. Therefore, the profiling result for CD55 serves as a good example to support the linear correlation between the expression level and Raman intensity. Nevertheless, we admit that extensive validation is required in the future to firmly establish the correlation between the intensity and expression level.

The curves in Fig. S6 were averaged Raman spectral from multiple cells, the sample size has been specified in the figure caption.

To clear potential confusion, we've modified the caption of Figure S6 as follows:

Figure S6. The specificity and semi-quantitative measurement of seven-colored Rdots conjugates. Cells were stained with targeted Rdots conjugates, respectively. (a) Rdot2218-Nucleolin stained positive cell HeLa and negative cell Jurkat, averaged over 105 and 88 cells, respectively; (b) Rdot2194-EpCAM stained positive cell SKBR3 and negative cell Jurkat, averaged over 112 and 140 cells, respectively; (c) Rdot2175-MUC1 stained positive cell HeLa and negative cell Jurkat, averaged over 75 and 80 cells, respectively; (d) Rdot2153-CD55 stained SKBR3 and Jurkat cells with moderate expression, averaged over 155 and 77 cells, respectively. According to the human protein atlas database, the normalized expression of CD55 is 1.5 for Jurkat cell and 10.1 for SKBR3 cell. Consistent with the expression levels reported, the peak intensity of SKBR3 cells is about 5 times larger than Jurkat cells, indicating a linear correlation between the Raman intensity and expression level. (e) Rdot2133-EGFR stained positive cell SKBR3 and negative cell Jurkat, averaged over 151 and 140 cells, respectively; (f) Rdot2079-CD44 stained positive cell HeLa and negative cell Jurkat, averaged over 124 and 83 cells, respectively; (g) Rdot2052-HER2 stained positive cell SKBR3 and negative cell Jurkat, averaged over 92 and 86 cells, respectively.

- Fig. S7 is missing sample size, and statistical significance (or absence of it) should be indicated.

Response:

We appreciate the suggestion from the reviewer, and have added sample size and statistical significance to Figure S7 (updated Figure S9).

- Page 7: some characters have encoding problems, I suppose it is supposed to be "Amide I" and "Amide III"

Response:

The Roman numbers were corrupted during PDF conversion, it should be "Amide I" and "Amide II". We have corrected them in the revision.

- Page 9: I agree that single-cell measurements is far more informative than average populations, but as this technique mostly compares with fluorescence and mass cytometry, which techniques are the authors referring to?

Response:

We are not exactly sure about the reviewer's question. Based on the literature, both the fluorescence cytometry and mass cytometry have reached single-cell level and thus have been used in single-cell profiling³⁻⁵. In this regard, it's fair to compare our single-cell Raman profiling with both fluorescence cytometry and mass cytometry.

- Page 11: The authors claim that it would be straightforward to extend that measurement approach to CARS and SRS to improve throughput, but on the other hand, due to the very small excitation volume in nonlinear optics, whole-cell spectra would have to be recorded in a completely different manner, which could very well reduce throughput as well. A more detailed discussion would be required to make such a claim.

Response:

We agree with the reviewer that, when coupled with SRS/CARS, the whole-cell information cannot be integrated into a single acquisition due to the small excitation volume used in SRS/CARS. However, a satisfactory throughput is still available with high-speed image scanning. Actually, this technique has been realized in some literature, which has been cited in our original manuscript.

For example, in Nitta, Nao, et al. *Nature communications* 11.1 (2020): 1-16, the SRS microscope employs a parallel detection system with a line-focusing optics and a 24-channel photodetector array for a quick cell scanning. Specifically, the synchronized pump and Stokes beams will pass through a beam shaper that generates an elliptically shaped beam to form a focal spot with a size of $24 \times 1 \mu\text{m}^2$ to illuminate a line of cell sample. The transmitted pump beam is then detected by a vertically aligned photodetector array. When equipped with a high-speed flow setup, a fast 2D SRS image of cells is digitally constructed by stacking the 1D SRS lines in the flow direction. Overall, this technology enables real-time SRS-image of single live cells with a throughput of up to ~100 events per second. Besides, the fast wavelength-switchable laser enables the high-speed multicolor SRS image acquisition. Another example: Suzuki, Yuta, et al. *Proceedings of the National Academy of Sciences* 116.32 (2019): 15842-15848. The cell flow is pumped at a flow speed of 2 cm/s at the center of the microchannel. For 2D SRS imaging, a resonant galvanometric scanner generates sinusoidal motion of the focal spot at 12 kHz, ensuring a throughput of up to ~140 cells/s.

When coupled with our Raman probe set, it's possible to realize multiplexed SRS-imaging of single live cells at high throughput. Compared with whole-cell profiling, single-cell imaging-based analysis additionally provides intracellular spatial distribution and morphological features of cells, which can add extra information to cell phenotyping. We hope these examples can help clarify the reviewer's concern, and some technical discussion has been added to the final discussion:

By virtue of nonlinear amplification, as well as the quick parallel detection and scanning system, SRS microscope enables real-time SRS-image of single live cells with a throughput of up to ~140 events per second. Moreover, a flow setup equipped with a sorting feature would facilitate downstream cell sorting.

Reference

1. Itzhak, D. N.; Tyanova, S.; Cox, J.; Borner, G. H., Global, quantitative and dynamic mapping of protein subcellular localization. *Elife* **2016**, *5*.
2. Perfetto, S. P., Pratip K. Chattopadhyay, and Mario Roederer, Seventeen-colour flow cytometry: unravelling the immune system. *Nature Reviews Immunology* **2004**, *4* (8), 648-655.
3. Spitzer, M. H.; Nolan, G. P., Mass Cytometry: Single Cells, Many Features. *Cell* **2016**, *165* (4), 780-91.
4. Bendall, S. C.; Nolan, G. P., From single cells to deep phenotypes in cancer. *Nat Biotechnol* **2012**, *30* (7), 639-47.
5. Brummelman, J.; Haftmann, C.; Nunez, N. G.; Alvisi, G.; Mazza, E. M. C.; Becher, B.; Lugli, E., Development, application and computational analysis of high-dimensional fluorescent antibody panels for single-cell flow cytometry. *Nat Protoc* **2019**, *14* (7), 1946-1969.

REVIEWERS' COMMENTS

Reviewer #1 (Remarks to the Author):

The authors have addressed all my comments. I recommend acceptance and would like to congratulate them on the nice work!

Reviewer #3 (Remarks to the Author):

In their revised version of their manuscript, the authors have addressed satisfactorily all concerns raised by the reviewers, making it suitable for publication.